# Interleaved Gibbs Diffusion: Generating Discrete-Continuous Data with Implicit Constraints

## Abstract

We introduce Interleaved Gibbs Diffusion (IGD), a novel generative modeling framework for discrete-continuous data, focusing on problems with important, implicit and unspecified constraints in the data. Most prior works on discrete and discrete-continuous diffusion assume a factorized denoising distribution, which can hinder the modeling of strong dependencies between random variables in such problems. We empirically demonstrate a significant improvement in 3-SAT performance out of the box by switching to a Gibbs-sampling style discrete diffusion model which does not assume factorizability. Motivated by this, we introduce IGD which generalizes discrete time Gibbs sampling type Markov chain for the case of discrete-continuous generation. IGD allows for seamless integration between discrete and continuous denoisers while theoretically guaranteeing exact reversal of a suitable forward process. Further, it provides flexibility in the choice of denoisers, allows conditional generation via state-space doubling and inference time refinement. Empirical evaluations on three challenging generation tasks - molecule structures, layouts and tabular data - demonstrate state-of-the-art performance. Notably, IGD achieves state-of-the-art results without relying on domain-specific inductive biases like equivariant diffusion or auxiliary losses. We explore a wide range of modeling, and interleaving strategies along with hyperparameters in each of these problems.

## 1 Introduction

Autoregressive models have been highly successful at modeling languages in a token by token fashion. While finetuned autoregressive (AR) models can produce realistic texts and maintain lengthy human-like conversations, they are known to fail at simple planning and reasoning tasks. One hypothesis is that AR generation is not suited for generating tokens where non-trivial constraints have to be satisfied (Zhang et al., 2023a). There have been efforts such as Chain-of-Thought prompting (Wei et al., 2022) and O1 (OpenAI, 2024) which force the model to "think over" the solution in many steps before answering.

Diffusion models, another class of generative models, start with pure noise and slowly denoise to obtain a sample from the desired distribution (Ho et al., 2020; Song et al., 2021). While its outstanding applications have been in the context of generating images (i.e, continuous data) (Saharia et al., 2022; Rombach et al., 2022), it has been successfully extended to discrete data (Austin et al., 2021; Lou et al., 2024).

Most data distributions considered for generative modeling have certain important constraints which need to be captured. These constraints are inherent or *implicit* if we do not algorithmically or systematically describe these constraints to the model other than presenting data in which these are satisfied. For example, in the language modeling task, the constraint is that generated sentences have to make grammatical sense. In this work, we use the term generative modeling with implicit constraints in the following sense:

- The learning algorithm does not provide the model with any information regarding the constraints during training, except presenting data points where the constraints are satisfied. The effectiveness of the generative model in capturing the implicit constraints is evaluated through direct/ proxy metrics only after inference. This design is necessary in cases where the implicit constraints may not be

easy to capture explicitly - for instance, in molecule generation, a molecule may be required to be stable (which would in turn mean constraints on bond lengths, bond angles and so on) and in layout generation, a layout may be required to be aesthetically pleasing (which is difficult to quantify). A more detailed exposition of the constraints and metrics in the tasks we consider in this work are given in Appendix B. Therefore, our objective is fundamentally different from the generative modeling works (Fishman et al., 2023; Christopher et al., 2024; Bouvier et al., 2025) which explicitly impose constraints algorithmically or works which make specific design choices (such as chain-of-thought) to better allow constraint satisfaction (Pan et al., 2024).

- Language modeling is also a modeling task with implicit constraints. However, the causal ordering imposed by AR models is already effective for language generation and in general outperforms discrete diffusion models. On the other hand, for planning tasks such as layout generation (Inoue et al., 2023), molecule generation (Hoogeboom et al., 2022), 3SAT, SuDoKu (Ye et al., 2025) and Traveling Salesman Problem (Zhang et al., 2025), diffusion models outperform AR models. This is attributed to diffusion models not imposing a causal ordering. Thus, the entire set of generated tokens are parsed multiple times during denoising. In this work, we are specifically interested in such problems where AR modeling is ineffective.

Given this context, we first note that discrete diffusion models based on D3PM (Austin et al., 2021) as presented in prior works (Inoue et al., 2023; Ye et al., 2025) make a specific modeling choice - they assume that the denoising process samples from a *product distribution* of the tokens. Intuitively, this factorization assumption seems unreasonable particularly for generation of data with strong implicit constraints, where the tokens can be highly inter-dependent. Hence, it is reasonable to expect alternative proposals such as Concrete Score Matching (Meng et al., 2022a), SEDD (Lou et al., 2024), symmetric diffusion ((Zhang et al., 2025)) and Glauber Generative Model (GGM) (Varma et al., 2025), which *do not assume such a factorization*, to do better in such problems.

Similarly, current discrete-continuous diffusion models, i.e., models which can sample from distributions on sequences with both discrete tokens and continuous vectors, *assume factorizability* across elements (Hua et al., 2024; Levi et al., 2023). This modeling choice may not be appropriate for sampling from datasets where we it is necessary to accurately capture the implicit constraints between the generated elements. Such problems arise naturally in applications like Layout Generation (Levi et al., 2023), Molecule Generation (Hua et al., 2024) and Tabular Data Generation (Jolicoeur-Martineau et al., 2024). In this context, we ask the question: can we construct a discrete-continuous diffusion framework, which *does not* assume factorizability, to accurately sample from a dataset with implicit constraints?

We take inspiration from the Gibbs Sampling literature - Gibbs Sampler is a Markov chain, widely studied in Theoretical Computer Science, Statistical Physics, Bayesian Inference and Probability Theory (Geman & Geman, 1984; Turchin, 1971; Gelfand & Smith, 1990; Martinelli, 1999; Levin & Peres, 2017). It samples jointly distributed random variables by resampling one co-ordinate at a time from the accurate conditional distribution. While the original form is a Markov Chain Monte Carlo (MCMC) algorithm, Varma et al. (2025) considered a learned, time dependent variant for generative modeling over discrete spaces. We extend the principle of time dependent Gibbs sampler to discrete-continuous data, to obtain a framework which *does not assume factorizability*.

**Our Contributions:** We introduce an effective method to train a discrete-continuous diffusion model to solve generation problems with implicit constraints. The key contributions include:

1. As a motivating example, we consider the proto-typical constrained generation problem of 3-SAT (a purely discrete generation problem) and demonstrate that *exact reversal* with Gibbs Sampling style algorithm (Varma et al., 2025) can *outperform* generative models which assume *factorizability* (Ye et al., 2025) out of the box. We perform a detailed empirical study across varying number of constraints, and show that the difference is starker with larger number of constraints.

2. We then propose Interleaved Gibbs Diffusion (IGD), a framework for sampling from **discrete-continuous distributions**, **provably achieving exact reversal** with ideal denoisers. As far as

we know, this is the *first* discrete-continuous framework which does *not assume factorizability* of the denoising process. The framework supports *conditional sampling* via state space doubling (inspired by Levi et al. (2023)) and inference-time refinement via *ReDeNoise* (inspired by Meng et al. (2022b)).

3. Multiple strategies to obtain these ideal denoisers are then discussed - we establish that *denoisers can be trained using well-understood diffusion objectives.* In particular, we show that discrete denoisers can be trained by considering an appropriate **classification** problem and that continuous denoisers can be trained using **a conditional score matching** objective.

4. We demonstrate **state-of-the-art performance** in molecule generation, layout generation and tabular data generation without relying on specialized diffusion processes or domain-specific architectures. For each task, a detailed empirical study of various modeling choices is also provided in the Appendix.

**Organization:** We motivate the necessity of exact reversal in Section 2. The IGD framework, with continuous and discrete denoisers as black boxes, is described in Section 3, along with the inference-time refinement ReDeNoise algorithm and conditional sampling algorithm. Multiple recipes to design and train the black box denoisers are given in Section 4. Experimental results are presented in Section 6, Conclusion and Future Work in Section 7.

## 2 Benefits of Exact Reversal for Problems with Implicit Constraints

As noted in Section 1, multiple existing works in discrete and discrete-continuous diffusion *assume factorizability across tokens in the reverse process.* More formally, let $q_\theta(x^{(s)}|x^{(t)})$ denote the reverse process for a trained diffusion model. Note that $q_\theta(x^{(s)}|x^{(t)})$ gives the distribution of sequence $x^{(s)}$ at timestep $s$ given the sequence $x^{(t)}$ at timestep $t$. Further, let $x^{(s)} = (x_1^{(s)}, x_2^{(s)}, \ldots, x_N^{(s)})$ and $x^{(t)} = (x_1^{(t)}, x_2^{(t)}, \ldots, x_N^{(t)})$, where $N$ is the number of elements in the sequence. Then, *if we assume factorizability*, the reverse process can be written as:

$$q_\theta(x^{(s)}|x^{(t)}) = \prod_{n=1}^{N} q_\theta(x_n^{(s)}|x^{(t)}) \tag{1}$$

However, it is known in literature that this is exact only in the continuous time limit, i.e. $s \to t$ (Shi et al., 2024). In practice, for finite timesteps, equation 1 does not hold and hence the reverse process is *not exact.* In contrast, there are other frameworks, like Varma et al. (2025), which do not assume such factorization and hence allow *exact reversal even for finite timesteps.* To showcase the impact of this distinction in problems with strong constraints between tokens, we consider the Boolean Satisfiability problem (SAT).

In SAT, the task is to find a binary assignment (if it exists) to the variables of a given boolean expression that makes it evaluate to *True.* In particular, we focus on random 3-SAT, a well-studied variation of SAT. Here, the dataset consists of random 3-SAT instances along with a binary assignment which satisfies the expression. During inference, a random 3-SAT instance is given as input and the model is expected to generate a binary assignment which satisfies the expression. This is a case of **implicitly constrained generation** since the binary expression is constrained to satisfy the boolean expression but the training algorithm is expected to learn this without explicitly specifying it.

We mirror the setup used in Ye et al. (2025) (see Appendix J for details). Note that MGDM (Ye et al., 2025) uses a variant of absorbing state D3PM (Austin et al., 2021) and hence *assumes factorizability* in the reverse process as per equation 1. We then implement GGM (Varma et al., 2025), a discrete diffusion framework which *does not assume factorizability*, in this setup and compare the performance.

As see in Table 1, exact reversal results in up to **7%** **improvement out of the box**, with *larger improvements for cases with more constraints.* We study this up to 20 variables and establish **SoTA results on 3-SAT for diffusion models** (Appendix J). Motivated by this observation, we now focus on developing such a framework in the more general discrete-continuous case.

Table 1: **SAT:** Accuracy with increasing number of variables $n$. Separate model trained for each $n$

| Method | Params | $n = 5$ | $n = 7$ | $n = 9$ |
|--------|--------|---------|---------|---------|
| GPT-2 Scratch | 6M | 97.6 | 85.6 | 73.3 |
| MGDM | 6M | 100.0 | 95.9 | 87.0 |
| GGM | 6M | 100.0 | 98.0 | 94.5 |
|  | 85M | - | 99.9 | 99.9 |

Figure 1: **Interleaved Forward Process:** Sequential forward process of discrete tokens ($D_s$) and continuous vectors ($C_s$). Forward process occurs one element at a time, keeping other elements unchanged - the process for discrete and continuous elements are *interleaved across rounds.*

## 3 Interleaved Gibbs Diffusion

**Notation:** Let $\mathcal{X}$ be a finite set. Let $L_1, L_2 \in \mathbb{N} \cup \{0\}$. Define the sequence length $L$ as $L = L_1 + L_2$. Let $d_{L_1+1}, \ldots, d_L \in \mathbb{N}$ be the continuous dimensions. We let our state space to be $\mathcal{S}_{\mathbf{L}} = \mathcal{X}^{L_1} \times_{i=L_1+1}^{L} \mathbb{R}^{d_i}$. The elements of this set can be represented as a tuple/sequence of length $L$. Each position in this sequence corresponds to either a discrete element drawn from $\mathcal{X}$ or a continuous element drawn from a real vector space. The number of positions corresponding to discrete elements is $L_1$ and the number of positions corresponding to continuous elements is $L_2$. For any $s \in \mathcal{S}_{\mathbf{L}}$, let $s_i$ denote the element in $s$ at position $i$ in the tuple. Note that $s_i$ is a discrete token from the set $\mathcal{X}$ if $i \leq L_1$ and it is a continuous vector sampled from $\mathbb{R}^{d_i}$ if $L_1 < i \leq L$. Let $s_{-i}$ denote the tuple of length $L - 1$ obtained by removing the element at the $i^{\text{th}}$ position of $s$. We reserve the uppercase letter $S$ (and its variants with subscripts and superscripts) to denote random variables/vectors/sequences and lowercase $s$ to denote a corresponding realization/sample .

**Objective:** Given samples $s_1, \ldots, s_N$ from the target distribution $\pi$ over $\mathcal{S}_{\mathbf{L}}$, the task is to learn a model which can generate more samples approximately from $\pi$.

We now describe the Interleaved Gibbs Diffusion (IGD) framework for sampling from a target distribution $\pi$ over both discrete and continuous elements. Since the objective is to have exact reversal in finite time, we choose a Gibbs Sampling type Markov chain for defining the forward and backward process. In particular, this framework offers two salient features which enable seamless integration of discrete and continuous elements:

**Forward and reverse processes operate one element at a time:** This allows us to theoretically guarantee exact reversal of the forward process, provided we have access to ideal denoisers, *even for a sequence with both discrete and continuous elements* (Lemma 3.2).

**Forward process factorizes across elements:** This allows us to adapt known results in continuous diffusion *to estimate the ideal denoisers*. In particular, we show that *score matching can be used for each continuous vector* to learn the ideal denoisers (Lemma 4.1).

### 3.1 Forward Process

Let $s^{(0)}$ be a sample from target distribution $\pi$. The forward process applies a discrete time Markov chain to obtain the trajectory $\{s^{(t)}\}_{t=0}^{T}$, where $T$ is the total number of timesteps. Let $\{S^{(t)}\}_{t=0}^{T}$ denote the corresponding random sequences and $\{P_t\}_{t=0}^{T}$ denote the distributions of these random sequences. We refer to $t$ as *sequence time*; unless specified otherwise, *time* refers to sequence time.

**Noise Order:** By construction, forward process operates *only on one element* at any given time $t$. Let $i_t \in \{1, 2, \ldots, L\}$ denote the sequence position that undergoes forward process at time $t$. Intuitively, we construct the forward process to proceed in *rounds* - we make sure every position is visited exactly once in every round of forward process. To achieve this, we choose $T = rL$, where $r \in \mathbb{N}, r > 1$ denotes the total number of rounds. For any round $r_1 \in \{1, \ldots, r\}$, we choose the positions $\{i_t\}_{t=(r_1-1)L}^{r_1 L}$ to be a fixed permutation of $\{1, 2, \ldots, L\}$ - we refer to this permutation as the *noise order*. For the rest of the discussion, it suffices to think of the forward process as operating on every element sequentially in a cyclic fashion, across multiple rounds.

**Interleaving:** Note that we consider the number of rounds to be greater than 1. After the first round, the forward process has operated on *all elements in the sequence*, i.e., by the second round, both discrete and continuous elements undergo forward process *irrespective of the noise order*. Hence, **across rounds**, the forward process of discrete and continuous elements are **interleaved**. The forward process in IGD is illustrated in Figure 1. We now describe how individual discrete and continuous elements undergo the forward process. First, **note that $\mathbf{s_{-i_t}^{(t+1)} = s_{-i_t}^{(t)}}$**, since **only the $i_t^{\mathbf{th}}$ element undergoes any change**. Now,

**If $s_{i_t}$ is discrete:** Following Varma et al. (2025), we consider token $\phi \notin \mathcal{X}$ and define a probability distribution $\Pi_t$ over $\mathcal{X} \cup \{\phi\}$. Note that $\Pi_t$, the *discrete noise schedule*, depends on the sequence time $t$. Then:

Sample $z_t \sim \Pi_t$ *independent of* $S^{(t)}$. We define:

$$s_j^{(t+1)} = \begin{cases} z_t, & \text{if } j = i_t \text{ and } z_t \neq \phi \\ s_j^{(t)}, & \text{otherwise} \end{cases}$$

**If $s_{i_t}$ is continuous:** We perform $K_{i_t}^t$ steps of DDPM (Ho et al., 2020) style Gaussian noising. In particular, define $k \in \{0, \ldots, K_{i_t}^t\}$ to be *element time*. Let $\beta$ denote the continuous noise schedule: $\beta$ outputs a scalar given $(t, k)$ as the input. Define $s_{i_t}^{(t,0)} = s_{i_t}^{(t)}$ and $s_{i_t}^{(t+1)} = s_{i_t}^{(t,K_{i_t}^t)}$. For $0 \leq k \leq K_{i_t}^t - 1$:

$$s_{i_t}^{(t,k+1)} \sim \mathcal{N}\left(\left(\sqrt{1 - \beta(t,k)}\right) s_{i_t}^{(t,k)}, (\beta(t,k))\, \mathbf{I}\right)$$

Hence, **forward process of any element $s_{i_t}^{(t)}$ at time $t$ is independent of other elements**. This fact can be used to prove that the forward process converges to a product noise distribution:

**Lemma 3.1** (Informal, Mild extension of Lemma 1 in Varma et al. (2025))**.** *For appropriate noise schedules $\Pi_t$ and $\beta(t,k)$, as $T \to \infty$, $P_T$ converges to the product distribution:* $\Pi\left(\cdot | \mathcal{X}\right)^{L_1} \times_{i=L_1+1}^{L} \mathcal{N}\left(0, \mathbf{I}_{d_i}\right)$ .

Using independence of forward process, $s^{(t)}$ can be sampled directly from $s^{(0)}$ (see Appendix G). All proofs are given in Appendix A.

### 3.2 Reverse Denoising Process

Let $\hat{s}^{(T)}$ be a sample from $P_T$, the terminal distribution of the forward process. The reverse denoising process also applies a discrete time Markov chain to obtain the trajectory $\{\hat{s}^{(t)}\}_{t=T}^{0}$. Like the forward process, reverse process also *happens one element at a time* - the only difference is that $t$ now goes from $T$ to 0. Further, $i_t$ **is the same across forward and reverse processes**, i.e., the element undergoing reverse process at time $t$ is precisely the element which underwent forward process at time $t$. Hence, **it is clear that $\hat{s}_{-i_t}^{(t)} = \hat{s}_{-i_t}^{(t+1)}$ for all** $t$. Now, depending on whether $\hat{s}_{i_t}^{(t+1)}$ is discrete or continuous, we use an appropriate denoiser. Recall

that $S^{(t)}$ denotes the random sequence obtained through the forward process at time $t$. We then define the denoisers as follows:

**Discrete Denoiser**, denoted by `DiscDen`$(s, i_t, t)$, is a learned sampler which approximates a probability distribution over $\mathcal{X}$ given a sequence sample $s$, position $i_t$ and time $t$ as inputs. In particular, based on the chosen training strategy, `DiscDen` approximates one of the following distributions: $\mathbb{P}(S_{i_t}^{(t)}|S_{-i_t}^{(t+1)} = s_{-i_t})$ or $\mathbb{P}(S_{i_t}^{(t)}|S^{(t+1)} = s)$. Note that for sampling at time $t$, we set $s = \hat{s}^{(t+1)}$.

**Continuous Denoiser**, denoted by `ContDen`$(s, i_t, t)$, is a learned sampler which samples approximately from a probability density over $\mathbb{R}^{d_{i_t}}$ given a sequence sample $s$, position $i_t$ and time $t$ as inputs. In particular, `ContDen` samples approximately from the following density: $f(S_{i_t}^{(t)}|S^{(t+1)} = s)$. Again, for sampling at time $t$, we set $s = \hat{s}^{(t+1)}$.

The exact reverse process using these denoisers is given in Algorithm 1. Importantly, this reverse process **guarantees exact reversal** given ideal denoisers:

**Lemma 3.2.** *Assume $\hat{S}^{(T)} \sim P_T$ and assume we have access to ideal discrete and continuous denoisers. Then, $\hat{S}^{(0)}$ obtained after $T$ steps of reverse denoising process, will be such that $\hat{S}^{(0)} \sim \pi$.*

Note that unlike the forward process, reverse process is not factorizable. Further, note that *reversal is exact irrespective of the particular choice of decode order*, provided the denoisers are ideal.

**ReDeNoise:** Due to imperfect training, it is possible that the learned denoisers cause $\hat{s}^{(0)}$ to be sampled away from $\pi$. SDEdit algorithm (Meng et al., 2022b) starts with samples which are out-of-distribution, noises it partially and denoises it with the learned diffusion model to obtain samples which are closer to the target dataset. Works on predictor-corrector methods for both discrete and continuous diffusion (Song et al., 2021; Campbell et al., 2022; Lezama et al., 2022; Zhao et al., 2025) also make use of such noising with learned corrector models. Restart algorithm (Xu et al., 2023b) to reduce discretization errors has also been proposed for continuous diffusion. Inspired by these works, we introduce a predictor-corrector framework for discrete-continuous diffusion termed ReDeNoise - start from $\hat{s}^{(0)}$ (output of Algorithm 1), run the forward process for $r'$ rounds to obtain $\bar{s}^{(r'L)}$ and then the reverse processes for $r'$ rounds starting from $\bar{s}^{(r'L)}$ to obtain the sample $\hat{\bar{s}}^{(0)}$. $r'$ here is a hyperparameter: $r' = 0$ recovers Algorithm 1, while $r' > 0$ can help correct for errors at inference time. We theoretically analyze how ReDeNoise can help correct the errors accumulated in the initial stages of denoising (Appendix C) and demonstrate results empirically (Appendix L.3).

---

**Algorithm 1** Interleaved Gibbs Diffusion: Ideal Denoising

---

**Input:** $\hat{s}^T \sim P_T$, discrete denoiser `DiscDen`, continuous denoiser `ContDen`, denoise positions $\{i_t\}$
**Output:** $\hat{s}^0 \sim \pi$
1: **for** $t \in [T-1, T-2, \ldots, 0]$ **do**
2:     Set $\hat{s}_{-i_t}^{(t)} = \hat{s}_{-i_t}^{(t+1)}$
3:     **if** $\hat{s}_{i_t}^{(t)}$ is discrete **then**
4:         $\hat{s}_{i_t}^{(t)} \leftarrow$ `DiscDen`$(\hat{s}^{(t+1)}, i_t, t)$
5:     **else**
6:         $\hat{s}_{i_t}^{(t)} \leftarrow$ `ContDen`$(\hat{s}^{(t+1)}, i_t, t)$
7:     **end if**
8: **end for**

---

### 3.3 Conditional Generation

We can also perform conditional generation using this framework - i.e, generate a subset of elements in a sequence conditioned on the rest. We adopt the state-space doubling strategy, inspired by Levi et al. (2023). A binary mask vector is created indicating whether each element in the sequence is part of the conditioning or not; for vectors in $\mathbb{R}^d$, a mask is created for each element in the vector. During forward and reverse processes, the elements which are part of conditioning remain unchanged. Further, the mask is also fed in while training the denoisers to indicate conditioning.

# 4 Training the Denoisers

Having established the IGD framework, we now describe strategies to train the black box discrete and continuous denoisers. We use $g_\theta$ to denote a parameterized neural network - we use the same neural network (with appropriate slicing at the output layer) to train both discrete and continuous denoisers. Recall that $g_\theta$ should take $(\hat{s}^{(t+1)}, i_t, t)$ as input and should be trained to sample from an appropriate distribution/density as discussed in Section 3.2. For discrete inputs, $g_\theta$ outputs logits in the space $[0,1]^{|\mathcal{X}|}$ and for continuous inputs, $g_\theta$ outputs a vector in the space $\mathbb{R}^{d_{i_t}}$.

## 4.1 Training the Discrete Denoiser

### 4.1.1 Alternative 1: Learning $\mathbb{P}\left(S_{i_t}^{(t)} | S^{(t+1)} = \hat{s}^{(t+1)}\right)$

Since the vocabulary has $|\mathcal{X}|$ discrete tokens, we require $g_\theta$ to predict $|\mathcal{X}|$ logits (i.e, a $|\mathcal{X}|$-**ary classification problem**). Given samples $(s^{(t)}, s^{(t+1)})$ from the forward process, we train $g_\theta$ by minimizing the cross-entropy loss: $\mathcal{L}_{CE}\left(\theta; s^{(t+1)}, i_t, t\right) = -\log\left(g_\theta^{s_{i_t}}\left(s^{(t+1)}, i_t, t\right)\right)$ where $g_\theta^{s_{i_t}}$ denotes the logit corresponding to token $s_{i_t}^{(t)}$.

### 4.1.2 Alternative 2: Learning $\mathbb{P}\left(S_{i_t}^{(t)} | S_{-i_t}^{(t+1)} = \hat{s}_{-i_t}^{(t+1)}\right)$

While this may also seem like a $|\mathcal{X}|$-ary classification problem, results from Varma et al. (2025) show that this can be reduced to a **binary classification problem** (see Appendix D for more details). Given samples $s^{(t)}, s^{(t+1)}$ and $z_t$ from the forward process (defined in Section 3.1), we minimize the binary cross-entropy loss:$\mathcal{L}_{BCE}\left(\theta; s_{-i_t}^{(t+1)}, i_t, t\right) = -\mathbf{1}_{z_t \neq \phi}\log\left(g_\theta^{s_{i_t}}\left(s_{-i_t}^{(t+1)}, i_t, t\right)\right) - \mathbf{1}_{z_t = \phi}\log\left(1 - g_\theta^x\left(s_{-i_t}^{(t+1)}, t\right)\right)$ where $g_\theta^{s_{i_t}}(\cdot)$ denotes the logit corresponding to token $s_{i_t}^{(t)}$.

Preliminary experiments (Appendix K.4) gave better results with the binary classification loss; hence we use *binary classification* for training the discrete denoiser.

## 4.2 Training the Continuous Denoiser

The objective is to learn the density $f(S_{i_t}^{(t)} | S^{(t+1)} = \hat{s}^{(t+1)})$. Mirroring the forward process, we do this through $K_{i_t}^t$ steps - if we set $\hat{s}^{(t, K_{i_t}^t)} = \hat{s}^{(t+1)}$ and $\hat{s}^{(t)} = \hat{s}^{(t,0)}$, it suffices to learn $f(S_{i_t}^{(t,k)} | S^{(t,k+1)} = \hat{s}^{(t,k+1)})$ for $0 \leq k \leq K_{i_t}^t - 1$.

Adapting prior works Ho et al. (2020); Song et al. (2021) to our setup, it can be seen that learning the density $f(S_{i_t}^{(t,k)} | S^{(t,k+1)} = \hat{s}^{(t,k+1)})$, for the given conditioning $\hat{s}^{(t,k+1)}$, can be reduced to learning the conditional score $\nabla \log f(S_{i_t}^{(t,k+1)} | S_{-i_t}^{(t,k+1)} = \hat{s}_{-i_t}^{(t,k+1)})$ (see Appendix E for exact details). The objective then is to estimate this conditional score. Towards this, let the random vector $\epsilon^{(t,k+1)}$ denote the effective cumulative noise added to $S_{i_t}^{(0)}$ to obtain $S_{i_t}^{(t,k+1)}$, i.e.

$$\epsilon^{(t,k+1)} = \frac{S_{i_t}^{(t,k+1)} - \sqrt{\bar{\alpha}(t,k+1)}S_{i_t}^{(0)}}{\sqrt{1 - \bar{\alpha}(t,k+1)}} \tag{2}$$

where $\bar{\alpha}$ is a cumulative noise schedule obtained from $\beta$ (see Appendix F). Then:

**Lemma 4.1.** *Under the considered forward process where noising occurs independently, we have:*

$$\nabla \log f(S_{i_t}^{(t,k+1)} | S_{-i_t}^{(t,k+1)} = \hat{s}_{-i_t}^{(t,k+1)}) = -\frac{1}{\sqrt{1 - \bar{\alpha}(t,k+1)}}\mathbb{E}\left[\epsilon^{(t,k+1)} | S^{(t,k+1)} = \hat{s}^{(t,k+1)}\right]$$

Lemma 4.1 adapts the well-known score matching result to our framework: the key difference is that the score here has *a dynamic conditioning*. However, by utilizing the fact that forward process factorizes across elements, it is shown that the *conditional score estimation* can still be reduced to *cumulative noise*

*estimation.* Hence, it is sufficient to estimate $\mathbb{E}\left[\epsilon^{(t,k+1)}|\hat{s}^{(t+1)}\right]$ - this is done by minimizing the regression loss $\left\|\epsilon^{(t,k+1)} - g_\theta\left(s^{(t,k+1)}, i_t, t, k\right)\right\|_2^2$.

**Note:** A detailed description of the exact training and inference algorithms we use is given in Appendix H.. Apart from DDPM (an SDE based method), we also evaluated DDIM (an ODE based method). However, preliminary results (Appendix L.3) indicated that DDPM performs better.

## 5    Model Architecture

Inspired by Peebles & Xie (2023), we use a transformer-based architecture closely resembling Diffusion Transformers (DiTs). Since DiT is designed for handling discrete tokens, we modify the architecture slightly to accommodate continuous vectors as well. Along with discrete token and time embeddings, projections of continuous vectors and their corresponding continuous time embeddings are also passed into the transformer blocks. Further, both discrete and continuous time information is incorporated into adaptive layer normalization (Xu et al., 2019). Exact details as well as diagrams are given in Appendix I.

## 6    Experiments

We evaluate the IGD framework on three different discrete-continuous generation tasks: Layout Generation, Molecule Generation, and Tabular Data Generation. Additional information regarding framework design (choices of $T, \{i_t\}, \beta_t, \Pi_t, K_{i_t}^t$), baselines, training, sampling step comparisons, architecture and ablations are provided in Appendices K, L and M.

### 6.1    Layout Generation

#### 6.1.1    Background

We aim to generate arrangements of UI elements (e.g., buttons, text blocks) or document components (e.g., titles, figures, tables) that satisfy both functional requirements and aesthetic principles. This problem is important in graphic design and interface prototyping. Formally, each layout is a set of $N$ elements $\{e_i\}_{i=1}^N$. Each element $e_i$ is represented by a discrete category $t_i \in \mathbb{N}$ and a continuous bounding box vector $\mathbf{p}_i \in \mathbb{R}^4$. We use the parameterization $\mathbf{p}_i = [x_i, y_i, l_i, w_i]^\top$, where $(x_i, y_i)$ represents the upper-left corner of the bounding box, and $(l_i, w_i)$ its length and width, respectively.

#### 6.1.2    Experimental Setup

We adopt a setup similar to Guerreiro et al. (2025) for standardized comparison to existing layout generation methods.

**Datasets:** We evaluate our method on two popular layout generation datasets:

1. PubLayNet (Zhong et al., 2019): Layouts of scientific documents annotated with 5 element categories.
2. RICO (Deka et al., 2017): User-interface (UI) layouts with 25 element categories.

Following prior works Jiang et al. (2023); Zhang et al. (2023b), layouts containing more than 20 elements are discarded.

**Evaluation metrics:** Following previous works Inoue et al. (2023); Chen et al. (2024), we adopt two metrics described below:

1. Frechet Inception Distance (**FID**) (Heusel et al., 2017): Measures distance between generated and real data distributions by comparing features extracted from a neural network. For FID calculation, we use the feature space from the same network with identical weights as in (Zhang et al., 2023b).
2. Maximum Intersection over Union (**mIoU**) (Kikuchi et al., 2021): Calculates the maximum IoU between bounding boxes of generated layouts and real data layouts with same element categories.

Results on additional evaluation metrics (Alignment and Overlap) are presented in Appendix K.1.

Table 2: **Layout Generation:** Quantitative results on the RICO and PubLayNet datasets. Refer to section 6.1.2 for details on evaluation tasks and metrics. Baseline metrics are taken from Inoue et al. (2023)

| Method | RICO | | | | | | PubLayNet | | | | | |
|---|---|---|---|---|---|---|---|---|---|---|---|---|
| | Unconditioned | | Category Conditioned | | Category+Size Conditioned | | Unconditioned | | Category Conditioned | | Category+Size Conditioned | |
| | FID ↓ | mIoU ↑ | FID ↓ | mIoU ↑ | FID ↓ | mIoU ↑ | FID ↓ | mIoU ↑ | FID ↓ | mIoU ↑ | FID ↓ | mIoU ↑ |
| LayoutTransformer | 24.32 | 0.587 | - | - | - | - | 30.05 | 0.359 | - | - | - | - |
| LayoutFormer++ | 20.20 | **0.634** | 2.48 | 0.377 | - | - | 47.08 | 0.401 | 10.15 | 0.333 | - | - |
| NDN-none | - | - | 13.76 | 0.350 | - | - | - | - | 35.67 | 0.310 | - | - |
| LayoutDM | 4.43 | 0.582 | 2.39 | 0.341 | 1.76 | 0.424 | 36.85 | 0.382 | 39.12 | 0.348 | 29.91 | 0.436 |
| DLT | 13.02 | 0.566 | 6.64 | 0.326 | 6.27 | 0.424 | 12.70 | **0.431** | 7.09 | 0.349 | 5.35 | 0.426 |
| LayoutDiffusion | 2.49 | 0.620 | 1.56 | 0.345 | - | - | 8.63 | 0.417 | 3.73 | 0.343 | - | - |
| LayoutFlow | **2.37** | 0.570 | 1.48 | 0.322 | 1.03 | 0.470 | 8.87 | 0.424 | **3.66** | 0.350 | 1.26 | 0.454 |
| Ours | 2.54 | 0.594 | **1.06** | **0.385** | **0.96** | **0.524** | **8.32** | 0.419 | 4.08 | **0.402** | **0.886** | **0.553** |

**Tasks:** Results are presented on three common layout generation tasks:

1. *Unconditioned* Generation: No constraints on elements categories or sizes. Element categories, positions and sizes of elements should be generated.
2. *Category-Conditioned* Generation: Element categories that should be present in the layout are specified. Positions and sizes of these elements should be generated.
3. *Category + Size-Conditioned* Generation: Both element categories and sizes of elements that should be present in the layout are specified. Only poisitions of these elements should be generated.

**Baselines:** Diffusion-based approaches include: LayoutDM (Inoue et al., 2023), which applies discrete diffusion to handle element categories and positions; LayoutDiffusion (Zhang et al., 2023b), employing iterative refinement with tailored noise schedules for layout attributes; and DLT (Levi et al., 2023), a hybrid model separating element categories and coordinates into distinct diffusion processes. Flow-based: LayoutFlow (Guerreiro et al., 2025) leverages trajectory learning for efficient sampling. Non-diffusion baselines comprise: LayoutTransformer (Gupta et al., 2021) (autoregressive sequence generation), LayoutFormer++ (Jiang et al., 2023) (serializes constraints into token sequences for conditional generation), and NDN-none (Lee et al., 2020) (adversarial training without constraints).

### 6.1.3 Results

Results are given in Table 2. On RICO, *we outperform all baselines* in category-conditioned and category+size-conditioned generation, with competitive performance on unconditioned generation. On PubLayNet, we achieve the best FID in unconditioned and category+size-conditioned generation.

Notably, **IGD outperforms DLT, a discrete-continuous diffusion model which assumes factorizability of the reverse process**, by a significant margin, demonstrating the benefits of exact reversal. Moreover, note that majority of the baselines use inductive biases, such as specialized diffusion processes (Inoue et al., 2023; Zhang et al., 2023b) and auxiliary losses (Guerreiro et al., 2025) - *we achieve SoTA results without any such modifications.*

## 6.2 Molecule Generation

### 6.2.1 Background

Molecule generation aims to synthesize new valid molecular structures from a distribution learned through samples - diffusion-based methods have shown strong capabilities in generating discrete atomic types and their corresponding 3D positions. We represent a molecule with $n$ atoms by $\{z_i, \mathbf{p}_i\}_{i=1}^{n}$, where $z_i \in \mathbb{N}$ is the atom's atomic number and $\mathbf{p}_i \in \mathbb{R}^3$ is the position.

Table 3: **Molecule Generation and Tabular Data Generation:** Quantitative results on QM9 molecule generation and average results across 5 tabular datasets. For molecule generation, we report mean (standard deviation) across 3 runs, each with 10K generated samples. Refer to section 6.2.2 and 6.3.2 for details on evaluation metrics. Baseline results taken from Hua et al. (2024) and Akazan et al. (2024) respectively.

| Molecule Generation | | | | |
|---|---|---|---|---|
| Method | Atom stable (%) | Mol stable (%) | Validity (%) | Uniqueness (%) |
| E-NF | 85.0 | 4.9 | 40.2 | 39.4 |
| G-Schnet | 95.7 | 68.1 | 85.5 | 80.3 |
| GDM | 97.6 | 71.6 | 90.4 | 89.5 |
| EDM | $98.7 \pm 0.1$ | $82.0 \pm 0.4$ | $91.9 \pm 0.5$ | $90.7 \pm 0.6$ |
| DiGress | $98.1 \pm 0.3$ | $79.8 \pm 5.6$ | **95.4** $\pm 1.1$ | $97.6 \pm 0.4$ |
| GeoLDM | **98.9**$\pm 0.1$ | $89.4 \pm 0.5$ | $93.8 \pm 0.4$ | $92.7 \pm 0.5$ |
| MUDiff | $98.8 \pm 0.2$ | $89.9 \pm 1.1$ | $95.3 \pm 1.5$ | **99.1** $\pm 0.5$ |
| Ours | **98.9**$\pm 0.03$ | **90.5** $\pm 0.15$ | **95.4** $\pm 0.2$ | $95.6 \pm 0.1$ |
| Data | 99.0 | 95.2 | 99.3 | 100.0 |

| Tabular Data Generation | | | | |
|---|---|---|---|---|
| Models | $W_{tr} \downarrow$ | $W_{te} \downarrow$ | $F_1^{gen} \uparrow$ | $F_1^{aug} \uparrow$ |
| HS3F-Euler | 0.596 | 1.321 | 0.763 | 0.787 |
| CS3F-Euler | 0.926 | 1.473 | 0.709 | 0.755 |
| HS3F-Rg4 | **0.584** | 1.313 | 0.747 | 0.756 |
| CS3F-Rg4 | 1.448 | 1.780 | 0.637 | 0.707 |
| ForestFlow | 1.064 | 1.461 | 0.703 | 0.747 |
| Ours | 0.593 | **1.292** | **0.778** | **0.794** |

### 6.2.2 Experimental Setup

We closely follow the methodology used in prior works Hua et al. (2024) and Hoogeboom et al. (2022) for 3D molecule generation.

**Datasets:** We evaluate on the popular QM9 benchmark (Ramakrishnan et al., 2014) which contains organic molecules with up to 29 atoms and their 3D coordinates. We adopt the standard 100K/18K/13K train/val/test split as in prior works. We generate all atoms, including hydrogen, since this is a harder task.

**Evaluation metrics:** We adopt four metrics following prior works Hua et al. (2024) and Hoogeboom et al. (2022):

1. **Atom Stability**: The fraction of atoms that satisfy their valency.
2. **Molecule Stability**: The fraction of molecules where all atoms are stable.
3. **Validity**: RDKit-based (Landrum et al., 2006) molecular sanitization checks, as in Hoogeboom et al. (2022).
4. **Uniqueness**: Fraction of unique and valid molecules.

**Baselines:** E-NF (equivariant normalizing flows) (Köhler et al., 2020) models molecular generation via invertible flow transformations. G-SchNet (Gebauer et al., 2019) employs an autoregressive architecture with rotational invariance. Diffusion-based approaches include EDM (Hoogeboom et al., 2022) (with SE(3)-equivariant network (Fuchs et al., 2020)) , GDM (Hoogeboom et al., 2022) (non-equivariant variant of EDM), and DiGress (Vignac et al., 2023a) (discrete diffusion for atoms/bonds without 3D geometry). GeoLDM (Xu et al., 2023a) leverages an equivariant latent diffusion process, while MUDiff (Hua et al., 2024) unifies discrete (atoms/bonds) and continuous (positions) diffusion with specialized attention blocks.

### 6.2.3 Results

Results are given in Table 3. IGD equals the best baselines in terms of atom stability and molecule validity while maintaining more than 95% unique samples among the valid molecules. Further, IGD achieves a molecule stability of 90.5%, *surpassing all baselines*. Notably, **IGD achieves competitive performance without relying on domain-specific inductive biases** such as equivariant diffusion or specialized attention blocks, which are crucial to the performance of the considered baselines.

## 6.3 Tabular data generation

### 6.3.1 Background

Tabular data generation refers to the task of generating synthetic tabular data using existing tabular datasets. Generating tabular data can help mitigate bias due to class imbalance (Juwara et al., 2024) and can enhance data security (Lee et al., 2021). In general, tabular datasets have feature heterogeneity, i.e., have both discrete and continuous features (Akazan et al., 2024). Each row in the tabular dataset is treated as a sample $s$. If the row has $N_D$ discrete features and $N_C$ continuous features, $s = (d_1, d_2, \cdots, d_{N_D}, \mathbf{c})$, where $\{d_1, \cdots, d_{N_D}\}$ are the discrete features and $\mathbf{c} \in \mathbb{R}^{N_c}$ is a continuous vector containing all continuous features.

### 6.3.2 Experimental setup

**Datasets:** Experiments are done on 5 tabular datasets (following Akazan et al. (2024)): blood-transfusion (Yeh, 2008), congress (con, 1987), car (Bohanec, 1988), tic-tac-toe (Aha, 1991) and glass (German, 1987). Table 3 reports mean results on these 5 datasets.

**Evaluation metrics:** Let us denote the train set as $D_{tr}$, the test set as $D_{te}$ and the generated data as $D_{gen}$. Further, $D_{aug}$ denotes the dataset obtained by combining $D_{tr}$ and $D_{gen}$. Then, we report:

1. $W_{tr}$ and $W_{te}$: 1-Wasserstein distance between ($D_{tr}$ and $D_{gen}$) and ($D_{te}$ and $D_{gen}$) respectively.
2. $F_1^{gen}$ and $F_1^{aug}$: Average $F_1$ scores on $D_{te}$ of 4 models trained on $D_{gen}$ and $D_{aug}$ respectively.

**Baselines:** ForestFlow (Jolicoeur-Martineau et al., 2024) uses conditional flow matching with XGBoost for learning the velocity field. Continuous Sequential Feature Forest Flow (CS3F) (Akazan et al., 2024) implements ForestFlow but with per-feature autoregressive generation - CS3F-Euler and CS3F-Rg4 indicate Euler and Runge-Kutta discretization respectively (for the ODE). Heterogeneous Sequential Feature Forest Flow (HS3F) is a generalization of CS3F which uses an XGBoost classifier for discrete feature generation. CS3F-Euler, CS3F-Rg4, HS3F-Euler and HS3F-Rg4 indicate Euler and Runge-Kutta discretizations respectively for CS3F and HS3F.

### 6.3.3 Results

Table 3 reports the results for tabular data generation. IGD achieves competitive performance with respect to $W_{tr}$ and **outperforms all other methods with respect to $W_{te}, F_1^{gen}$ and $F_1^{aug}$**, proving the effectiveness of our framework. Further, note that HS3F is a per-feature autoregressive version of the flow matching based method ForestFlow. In a broad sense, IGD can be thought of **as a rigorous multi-round extension** of HS3F, and hence could lead to better performance.

## 7 Conclusion, Limitations and Future Work

We propose IGD, a diffusion framework for sampling from discrete-continuous data with implicit constraints. We theoretically establish the exactness of this framework and empirically validate its effectiveness across multiple tasks through extensive experiments.

One limitation of the current work is that the noise order and noise schedules are chosen in an ad-hoc manner through experimentation - future work can explore developing more principled approaches for these choices. Further, the focus of the current work is on obtaining quality improvements. The compute/ time complexity of IGD at inference would depend on the chosen interleaving pattern - a detailed exposition of the exact cost associated with different choices is given in Appendices K.4, L.3. For instance, compared to the baselines, IGD requires fewer sampling steps for molecule generation but more for layout generation. Future work can thus explore architectural improvements for improving compute/time complexity for interleaving patterns which are expensive.

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

## A Proofs

### A.1 Lemma 3.1

**Exact Statement:** Denote the distribution of $S^{(t)}$ by $P_t$. Suppose $\Pi_t\left(\cdot|\mathcal{X}\right) = \Pi\left(\cdot|\mathcal{X}\right)$ for all $t$, $\Pi_t(\phi) \leq 1 - \epsilon$ for some $\epsilon > 0$ and $\lim_{T\to\infty} \sum_{t=0}^{T} \sum_{k=0}^{K_{i_t}^t} \log\left(1 - \beta(t,k)\right) = -\infty$. As $T \to \infty$, $P_T$ converges to the product distribution: $\Pi\left(\cdot|\mathcal{X}\right)^{L_1} \times_{i=L_1+1}^{L} \mathcal{N}\left(0, \mathbf{I}_{d_i}\right)$.

**Proof:** We closely follow the proof of Lemma 1 in Varma et al. (2025).

Note that the forward process for each element is independent of all other elements. Hence, it suffices to consider the forward process for each element separately.

Consider a discrete element. By assumption, the probability of not choosing $\phi$:

$$1 - \Pi_t(\phi) \geq \epsilon$$

where $\epsilon > 0$ for all. Further, when $\phi$ is not chosen at time $t$, then the distribution of the discrete token is $\Pi(\cdot|\mathcal{X})$ for all time $\geq t$ independent of other tokens. The probability of choosing only $\phi$ until time $t$ is at most $(1-\epsilon)^t$ and this goes to 0 as $t \to \infty$. Therefore with probability 1, asymptotically, every discrete element converges to the distribution $\Pi\left(\cdot|\mathcal{X}\right)$.

Consider a continuous vector at position $i$. From the definition of the forward process, we have:

$$s_i^{(t,k+1)} = \left(\sqrt{1 - \beta(t,k)}\right) s_i^{(t,k)} + \left(\sqrt{\beta(t,k)}\right)\epsilon \tag{3}$$

where $\epsilon \sim \mathcal{N}(0,\mathbf{I})$. Merging the Gaussians, we have:

$$s_{i_t}^{(t,k+1)} = \left(\sqrt{\bar{\alpha}(t,k)}\right) s_{i_t}^{(0)} + \left(\sqrt{1 - \bar{\alpha}(t,k)}\right)\epsilon$$

where:

$$\bar{\alpha}(t,k) = \prod_{t'=0}^{t-1}\prod_{k'=0}^{K_i^{t'}}\left(1 - \beta(t',k')\right)\prod_{k''=0}^{k}\left(1 - \beta(t,k'')\right)$$

From the assumption $\lim_{T\to\infty}\sum_{t=0}^{T}\sum_{k=0}^{K_{i_t}^t}\log\left(1 - \beta(t,k)\right) = -\infty$, we have $\lim_{t\to\infty}\bar{\alpha}(t,k) = 0$ and hence the continuous vector will converge to an independent Gaussian with variance 1 per continuous dimension.

## A.2 Lemma 3.2

**Statement:** Assume $\hat{S}^{(T)} \sim P_T$ and assume we have access to ideal discrete and continuous denoisers. Then, $\hat{S}^{(0)}$ obtained after $T$ steps of reverse denoising process, will be such that $\hat{S}^{(0)} \sim \pi$.

**Proof:** Recall that $S^{(t)}$ denotes the random vector corresponding to sequence time $t$ of the forward process, $P_t$ denotes the probability measure of $S^{(t)}$ over $\mathcal{S}_{\mathbf{L}}$, $\hat{S}^{(t)}$ denotes the random vector corresponding to sequence time $t$ of the reverse process and $\hat{P}_t$ denote the probability measure of $\hat{s}^{(t)}$ over $\mathcal{S}_{\mathbf{L}}$.

We now prove the lemma by induction. Assume that $\hat{S}^{(t+1)} \stackrel{d}{=} S^{(t+1)}$, i.e., $P_{t+1} = \hat{P}_{t+1}$. Consider a measurable set $\mathcal{A}$ such that $\mathcal{A} \subseteq \mathcal{S}_L$ . Let $y \sim \hat{P}_{t+1}$. From the measure decomposition theorem, we have:

$$\mathbb{P}(\hat{S}^{(t)} \in \mathcal{A}) = \int_y \mathbb{P}\left(\hat{S}^{(t)} \in \mathcal{A}|\hat{S}^{(t+1)} = y\right) d\hat{P}_{t+1}(y)$$

From the induction assumption, we can rewrite this as:

$$\mathbb{P}(\hat{S}^{(t)} \in \mathcal{A}) = \int_y \mathbb{P}\left(\hat{S}^{(t)} \in \mathcal{A}|\hat{S}^{(t+1)} = y\right) dP_{t+1}(y)$$

From the definition of the reverse process, we know that $\hat{s}^{(t)}_{-i_t} = \hat{s}^{(t+1)}_{-i_t}$. Therefore, we have:

$$\mathbb{P}\left(\hat{S}^{(t)} \in \mathcal{A}|\hat{S}^{(t+1)} = y\right) = \mathbb{P}\left(\hat{S}^{(t)}_{i_t} \in \mathcal{A}_{-i_t}\left(y_{-i_t}\right)|\hat{S}^{(t+1)} = y\right)$$

where $\mathcal{A}_{-i_t}\left(y_{-i_t}\right) = \{x_{i_t} : x \in \mathcal{A}, x_{-i_t} = y_{-i_t}\}$. Depending on the reverse process chosen, we have:

$$\mathbb{P}\left(\hat{S}^{(t)}_{i_t} \in \mathcal{A}_{-i_t}\left(y_{-i_t}\right)|\hat{S}^{(t+1)} = y\right) = \mathbb{P}\left(S^{(t)}_{i_t} \in \mathcal{A}_{-i_t}\left(y_{-i_t}\right)|S^{(t+1)} = y\right) \quad \text{or}$$

$$\mathbb{P}\left(\hat{S}^{(t)}_{i_t} \in \mathcal{A}_{-i_t}\left(y_{-i_t}\right)|\hat{S}^{(t+1)} = y\right) = \mathbb{P}\left(S^{(t)}_{i_t} \in \mathcal{A}_{-i_t}\left(y_{-i_t}\right)|S^{(t+1)}_{-i_t} = y_{-i_t}\right)$$

Note that since we use the measure formalism, $\mathbb{P}\left(S^{(t)}_{i_t} \in \mathcal{A}_{-i_t}\left(y_{-i_t}\right)|S^{(t+1)} = y\right)$ handles both the discrete distribution $\mathbb{P}\left(S^{(t)}_{i_t}|S^{(t+1)} = s\right)$ and the continuous density $f\left(S^{(t)}_{i_t}|S^{(t+1)} = s\right)$ described in Section 3.2.

**Case 1:** $\mathbb{P}\left(\hat{S}^{(t)}_{i_t} \in \mathcal{A}_{-i_t}\left(y_{-i_t}\right)|\hat{S}^{(t+1)} = y\right) = \mathbb{P}\left(S^{(t)}_{i_t} \in \mathcal{A}_{-i_t}\left(y_{-i_t}\right)|S^{(t+1)} = y\right)$

We have:

$$\mathbb{P}\left(\hat{S}^{(t)} \in \mathcal{A}|\hat{S}^{(t+1)} = y\right) = \mathbb{P}\left(S^{(t)}_{i_t} \in \mathcal{A}_{-i_t}\left(y_{-i_t}\right)|S^{(t+1)} = y\right)$$

And hence:

$$\mathbb{P}(\hat{S}^{(t)} \in \mathcal{A}) = \int_y \mathbb{P}\left(S^{(t)}_{i_t} \in \mathcal{A}_{-i_t}\left(y_{-i_t}\right)|S^{(t+1)} = y\right) dP_{t+1}(y)$$

$$= \mathbb{P}(S^{(t)} \in \mathcal{A})$$

**Case 2:** $\mathbb{P}\left(\hat{S}^{(t)}_{i_t} \in \mathcal{A}_{-i_t}\left(y_{-i_t}\right)|\hat{S}^{(t+1)} = y\right) = \mathbb{P}\left(S^{(t)}_{i_t} \in \mathcal{A}_{-i_t}\left(y_{-i_t}\right)|S^{(t+1)}_{-i_t} = y_{-i_t}\right)$

We have:

$$\mathbb{P}\left(\hat{S}^{(t)} \in \mathcal{A}|\hat{S}^{(t+1)} = y\right) = \mathbb{P}\left(S^{(t)}_{i_t} \in \mathcal{A}_{-i_t}\left(y_{-i_t}\right)|S^{(t+1)}_{-i_t} = y_{-i_t}\right)$$

And hence:

$$\mathbb{P}(\hat{S}^{(t)} \in \mathcal{A}) = \int_y \mathbb{P}\left(S^{(t)}_{i_t} \in \mathcal{A}_{-i_t}\left(y_{-i_t}\right)|S^{(t+1)}_{-i_t} = y_{-i_t}\right) dP_{t+1}(y)$$

By measure decomposition theorem $P_{t+1}(y)$ is factorizable as:

$$P_{t+1}(y) = P_{t+1,-i_t}(y_{-i_t})P_{t+1,i_t}(y_{i_t}|y_{-i_t})$$

Therefore:

$$
\begin{aligned}
\mathbb{P}(\hat{S}^{(t)} \in \mathcal{A}) &= \int_y \mathbb{P}\left(S_{i_t}^{(t)} \in \mathcal{A}_{-i_t}(y_{-i_t}) \,|\, S_{-i_t}^{(t+1)} = y_{-i_t}\right) (dP_{t+1,-i_t}(y_{-i_t}))\,(dP_{t+1,i_t}(y_{i_t}|y_{-i_t})) \\
&= \int_{y_{-i_t}} \mathbb{P}\left(S_{i_t}^{(t)} \in \mathcal{A}_{-i_t}(y_{-i_t}) \,|\, S_{-i_t}^{(t+1)} = y_{-i_t}\right) (dP_{t+1,i_t}(y_{i_t}|y_{-i_t}))) \int_{y_{i_t}} (dP_{t+1,i_t}(y_{i_t})) \\
&= \mathbb{P}(S^{(t)} \in \mathcal{A})
\end{aligned}
$$

Hence, we have $\hat{S}^{(t)} \overset{d}{=} S^{(t)}$, i.e. $P_t = \hat{P}_t$. Therefore, by induction $\hat{P}_0 = \pi$, provided $\hat{P}_T = P_T$.

### A.3 Lemma 4.1

**Statement:** Under the considered forward process where noising occurs independently, we have:

$$\nabla \log f(S_{i_t}^{(t,k+1)} = \hat{s}_{i_t}^{(t,k+1)} | S_{-i_t}^{(t,k+1)} = \hat{s}_{-i_t}^{(t,k+1)}) = -\frac{1}{\sqrt{1 - \bar{\alpha}(t,k+1)}} \mathbb{E}\left[\epsilon^{(t,k+1)} | S^{(t,k+1)} = \hat{s}^{(t,k+1)}\right]$$

**Proof:** Let us split $S^{(t,k+1)} = \left[S_{i_t}^{(t,k+1)} \; S_{-i_t}^{(t,k+1)}\right]$.

$$\nabla \log q(S_{i_t}^{(t,k+1)} | S_{-i_t}^{(t,k+1)} = \hat{s}_{-i_t}^{(t,k+1)}) = \frac{\nabla q(S_{i_t}^{(t,k+1)} | S_{-i_t}^{(t,k+1)} = \hat{s}_{-i_t}^{(t,k+1)})}{q(S_{i_t}^{(t,k+1)} | S_{-i_t}^{(t,k+1)} = \hat{s}_{-i_t}^{(t,k+1)})}$$

$$= \frac{\nabla q(S_{i_t}^{(t,k+1)} | S_{-i_t}^{(t,k+1)} = \hat{s}_{-i_t}^{(t,k+1)}) q(S_{-i_t}^{(t,k+1)} = \hat{s}_{-i_t}^{(t,k+1)})}{q(S_{i_t}^{(t,k+1)} | S_{-i_t}^{(t,k+1)} = \hat{s}_{-i_t}^{(t,k+1)}) q(S_{-i_t}^{(t,k+1)} = \hat{s}_{-i_t}^{(t,k+1)})}$$

$$= \frac{\nabla q(S_{i_t}^{(t,k+1)}, S_{-i_t}^{(t,k+1)} = \hat{s}_{-i_t}^{(t,k+1)})}{q(S_{i_t}^{(t,k+1)}, S_{-i_t}^{(t,k+1)} = \hat{s}_{-i_t}^{(t,k+1)})}$$

$$= \frac{\nabla \int q(S_{i_t}^{(t,k+1)}, S_{-i_t}^{(t,k+1)} = \hat{s}_{-i_t}^{(t,k+1)} | S_{i_t}^{(0)}) q(S_{i_t}^{(0)}) dS_{i_t}^{(0)}}{q(S_{i_t}^{(t,k+1)}, S_{-i_t}^{(t,k+1)} = \hat{s}_{-i_t}^{(t,k+1)})}$$

$$\overset{(a)}{=} \frac{\nabla \int q(S_{i_t}^{(t,k+1)} | S_{i_t}^{(0)}) q(S_{-i_t}^{(t,k+1)} = \hat{s}_{-i_t}^{(t,k+1)} | S_{i_t}^{(0)}) q(S_{i_t}^{(0)}) dS_{i_t}^{(0)}}{q(S_{i_t}^{(t,k+1)}, S_{-i_t}^{(t,k+1)} = \hat{s}_{-i_t}^{(t,k+1)})}$$

$$\overset{(b)}{=} \frac{\int \frac{-\epsilon^{(t,k+1)}}{\sqrt{1-\bar{\alpha}(t,k+1)}} q(S_{i_t}^{(t,k+1)} | S_{i_t}^{(0)}) q(S_{-i_t}^{(t,k+1)} = \hat{s}_{-i_t}^{(t,k+1)} | S_{i_t}^{(0)}) q(S_{i_t}^{(0)}) dS_{i_t}^{(0)}}{q(S_{i_t}^{(t,k+1)}, S_{-i_t}^{(t,k+1)} = \hat{s}_{-i_t}^{(t,k+1)})}$$

$$\overset{(a)}{=} \frac{\int \frac{-\epsilon^{(t,k+1)}}{\sqrt{1-\bar{\alpha}(t,k+1)}} q(S_{i_t}^{(t,k+1)}, S_{-i_t}^{(t,k+1)} = \hat{s}_{-i_t}^{(t,k+1)}, S_{i_t}^{(0)}) dS_{i_t}^{(0)}}{q(S_{i_t}^{(t,k+1)}, S_{-i_t}^{(t,k+1)} = \hat{s}_{-i_t}^{(t,k+1)})}$$

$$= \int \frac{-\epsilon^{(t,k+1)}}{\sqrt{1 - \bar{\alpha}(t,k+1)}} q(S_{i_t}^{(0)} | S_{i_t}^{(t,k+1)}, S_{-i_t}^{(t,k+1)} = \hat{s}_{-i_t}^{(t,k+1)}) dS_{i_t}^{(0)} \tag{4}$$

In the RHS of the chain in equation 4, we observe that $S^{t,k+1}$ is being conditioned on. Given $S^{t,k+1}$, $\epsilon^{(t,K+1)}$ is a function only of $S_{i_t}^{(0)}$ from equation 2. Therefore, the above chain yields:

$$\nabla \log f(S_{i_t}^{(t,k+1)} = \hat{s}_{i_t}^{(t,k+1)} | S_{-i_t}^{(t,k+1)} = \hat{s}_{-i_t}^{(t,k+1)}) = -\frac{1}{\sqrt{1 - \bar{\alpha}(t,k+1)}} \mathbb{E}\left[\epsilon^{(t,k+1)} | S^{(t,k+1)} = \hat{s}^{(t,k+1)}\right]$$

This is the exactly $\frac{g_\theta(\cdot)}{\sqrt{1-\bar{\alpha}}}$ if the estimator is a perfect MMSE estimator.

Justifications:-
(a) Observe that conditioned on $S_{i_t}^{(0)}$, how $i_t$-th element is modified in the forward process is independent of all other elements. This gives rise to the conditional independence.
(b) We exchange the integral and the $\nabla$ operator. Let $q(x|y)$ be conditionally Gaussian, i.e. $x|y \sim \mathcal{N}(\sqrt{\bar{\alpha}}y; (1-\bar{\alpha}))$, then it is a property of the conditional Gaussian random variable that $\nabla_x q(x|y) = -\left(\frac{x - \sqrt{\bar{\alpha}}y}{1-\bar{\alpha}}\right) * q(x|y)$. Taking $x = S_{i_t}^{(t,k+1)}$ and $y = S_{i_t}^{(0)}$, we see that: $\nabla q(S_{i_t}^{(t,k+1)} | S_{i_t}^{(0)}) = \frac{-\epsilon^{(t,k+1)}}{\sqrt{1-\bar{\alpha}(t,k+1)}} * q(S_{i_t}^{(t,k+1)} | S_{i_t}^{(0)})$.

## B   Implicit Constraints in the Considered Datasets:

While exact modeling of distributions without assuming factorizability can be useful for any discrete-continuous modeling task in general, it can be particularly useful for modeling generation tasks with strong dependencies between the modeled random variables. Often, these dependencies are difficult to capture explicitly - this is what we refer to as implicit constraints. We now list some of these constraints for the tasks that we consider:

- **3SAT:** The implicit constraint dictates that the generated binary sequence should satisfy all the clauses. Note that this can be measured exactly.

- **Layout Generation:** The boxes should be aligned and their overlap should be small. These are measured by alignment and overlap metrics respectively. The layouts have to look "appealing" - this is hard to describe quantitatively. Hence, this is measured with widely used proxy metrics such as FID and MIoU.

- **Molecule Generation:** The positions and atom types should describe a stable, physically viable molecule. This is measured via molecule stability and validity, which check for the validity of properties such as valencies, bond angles and bond lengths.

- **Tabular Data Generation:** The constraints are different across different datasets. For example, in the tic-tac-toe dataset, the positions of the Xs and Os and the outcome of the game are constrained. In general, the values that the categorical/continuous features can take are heavily constrained by the target label - this is measured using F1 scores of classifiers trained on generated data as proxy metrics.

## C   Why ReDeNoise Helps:

The theoretical results in Lemma 3.2 show that if we learn the ideal denoiser, then the sampled distribution is accurate. However, we do not have access to the ideal denoiser due to finite amount of compute and data. To understand how ReDeNoise can help in this scenario, we first need to model how the error in the denoiser propagates into the error in the distribution of the generated samples. Thus, we first define the following notion of distance over the set of probability measures over $\mathcal{S}_\mathbf{L}$.

Recall the definition of the state space $\mathcal{S}_\mathbf{L} := \mathcal{X}^{L_1} \times_{i=L_1+1}^{L} \mathbb{R}^{d_i}$. Let us consider the following cost function between two elements of $s, s' \in \mathcal{S}_\mathbf{L}$: $D(s, s') := \sum_{i=1}^{L_1} \mathbb{1}(s_i \neq s'_i) + \sum_{i=L_1+1}^{L} \|s_i - s'_i\|^2$. This is the sum of $0-1$ distances of the discrete elements and the squared euclidean distances of the continuous elements. We consider the Wasserstein distance corresponding to the cost $D$, denoted as $\mathcal{W}_D$, defined below.

Let $P, Q$ be two probability measures over $\mathcal{S}_\mathbf{L}$ with finite second moments for the continuous elements. A coupling between $P, Q$ is a probability measure over $\mathcal{S}_\mathbf{L} \times \mathcal{S}_\mathbf{L}$ such that the marginal of the first co-ordinate is $P$ and the marginal of the second co-ordinate is $Q$. Let $\Gamma(P, Q)$ be the set of all couplings between $P$ and $Q$. Then,

$$\mathcal{W}_D^2(P, Q) := \inf_{\gamma \in \Gamma(P,Q)} \int D(s, s') d\gamma(s, s')$$

It can be easily shown via optimal transport theory that an optimal coupling, $\gamma^*$, exists Villani et al. (2008). We show that our forward process is a contraction of $\mathcal{W}_D^2$ after a whole round. For the sake of clarity of exposition, assume that Denoise order is given by the identity permutation. However, this can be easily extended to arbitrary orders. We also assume that the number of DDPM steps $K_{i_t}^t$ per noising is a constant $K$.

Define:

1. $\alpha_{r-1}^d := \min_{(r-1)L \leq t \leq (r-1)L+L_1-1} \mathbb{P}(Z_t \neq \phi)$

2. $\alpha_{r-1}^c := \min_{(r-1)L+L_1 \leq t \leq rL-1} 1 - \prod_{k=0}^{K-1}(1 - \beta(t, k))$

3. $\alpha_{r-1} = \min(\alpha_{r-1}^d, \alpha_{r-1}^c)$

**Lemma C.1.** *Given probability measures $\mu_0, \nu_0$ over $\mathcal{S}_\mathbf{L}$, let $\mu_t$ (resp. $\nu_t$) be the law of the forward process at time $t$ starting from $\mu_0$ (resp. $\nu_0$). Assume that $\mu_0, \nu_0$ have finite second moments.*

*Then,*
$$\mathcal{W}_D^2(\mu_{rL}, \nu_{rL}) \leq (1 - \alpha_{r-1})\mathcal{W}_D^2(\mu_{(r-1)L}, \nu_{(r-1)L}).$$

*Proof.* This proof follows from a direct coupling argument. Suppose $S^{(r-1)L} \sim \mu_{(r-1)L}$ and $\bar{S}^{(r-1)L} \sim \nu_{(r-1)L}$ be jointly distributed and optimally coupled with respect to the cost function $D$. We obtain $S^{rL} \sim \mu_{rL}$ and $\bar{S}^{rL} \sim \nu_{rL}$ from $S^{(r-1)L}$, $\bar{S}^{(r-1)L}$ by the same forward process as follows:

1. During a discrete noising timestep $t$, we draw $z_t \sim \Pi_t$ and update both $S^{(t)}$ and $\bar{S}^{(t)}$ with the same $z_t$.

2. During continuous noising timestep $t$, we use the same gaussian random variable to noise both $S^{(t)}$ and $\bar{S}^{(t)}$.

By definition of $\mathcal{W}_D$, it is clear that

$$\mathcal{W}_D^2(\mu_{rL}, \nu_{rL}) \leq \mathbb{E}D(S^{rL}, \bar{S}^{rL}) \tag{5}$$

Note that for $j \leq L_1$, if $(r-1)L \leq t < rL$ is the time such that $i_t = j$, $\mathbb{1}(S_j^{rL} \neq \bar{S}_j^{rL}) = \mathbb{1}(Z_t = \phi)\mathbb{1}(S_j^{(r-1)L} \neq \bar{S}_j^{(r-1)L})$, since, $Z_t$ is independent of $S^{(r-1)L}$ and $\bar{S}^{(r-1)L}$. Therefore,

$$\mathbb{E}\mathbb{1}(S_j^{rL} \neq \bar{S}_j^{rL}) \leq (1 - \alpha_{r-1})\mathbb{E}\mathbb{1}(S_j^{(r-1)L} \neq \bar{S}_j^{(r-1)L}) \tag{6}$$

Similarly, if $j > N_1$, our coupling ensures that:

$$S_j^{rL} - \bar{S}_j^{rL} = \prod_{k=0}^{K-1} \sqrt{1 - \beta(t,k)}(S_j^{(r-1)L} - \bar{S}_j^{(r-1)L})$$

Thus, we have:

$$\mathbb{E}\|S_j^{rL} - \bar{S}_j^{rL}\|^2 = (1 - \alpha_{r-1})\mathbb{E}\|S_j^{(r-1)L} - \bar{S}_j^{(r-1)L}\|^2 \tag{7}$$

Using Equations equation 6 and equation 7 in Equation equation 5 along with the definition of $D(,)$, we conclude the result. $\square$

**The Error Propagation Model:** Suppose $(P_t)_{t=0,...,T}$ be the law of the trajectory taken by the ideal denoiser plugged into Algorithm 1, starting from $P_T$. By Lemma 3.2, we know that $P_0$ is the exact target distribution. Similarly, let $(Q_t)_{t=0,...,T}$ be the law of the trajectory taken by the imperfect, learned denoiser plugged into Algorithm 1.

We make the following assumption: Suppose the law at time $rL$ during denoising is $\mu$ and we denoise for one round and obtain the law $\mu'$ at time $(r-1)L$ with the trained denoiser. Then,

$$\mathcal{W}_D^2(P_{(r-1)L}, \mu') \leq \underbrace{\mathcal{W}_D^2(P_{rL}, \mu)}_{\text{Error from } T \text{ to } rL} + \underbrace{\mathcal{E}_r}_{\text{Error in from } rL \text{ to } (r-1)L} \tag{8}$$

**Analyzing ReDeNoise:** Let $P_t$ and $Q_t$ be as defined above. Suppose $Q_L^{(1)}$ (resp. $P_L^{(1)}$) be obtained by the noising $Q_0$ (resp. $P_0$) by one round. Notice that by Lemma 3.2, we must have $P_L^{(1)} = P_L$ since the reverse process with the ideal denoiser perfectly reverses the forward process.

Now, by Lemma C.1, we have:

$$\mathcal{W}_D^2(P_L, Q_L^{(1)}) \leq (1 - \alpha_0)\mathcal{W}_D^2(P_0, Q_0)$$

Now suppose that $Q_0^{(1)}$ is obtained by denoising $Q_L^{(1)}$ for the last round. By Equation equation 8, we have:

$$\begin{aligned}
\mathcal{W}_D^2(P_0, Q_0^{(1)}) &\leq \mathcal{W}_D^2(P_L, Q_L^{(1)}) + \mathcal{E}_1 \\
&\leq (1 - \alpha_0)\mathcal{W}_D^2(P_0, Q_0) + \mathcal{E}_1
\end{aligned} \tag{9}$$

Then, whenever $\mathcal{W}_D^2(P_0, Q_0) \leq \frac{\mathcal{E}_1}{\alpha_0}$, we have: $\mathcal{W}_D^2(P_0, Q_0^{(1)}) \leq \mathcal{W}_D^2(P_0, Q_0)$, leading to an improvement in the sample quality as measured by $\mathcal{W}_D$.

Now, suppose we repeat ReDeNoise procedure $l$ times to obtain $Q_0^{(l)}$. That is, $Q_0^{(k+1)}$ is obtained by applying ReDeNoise to $Q_0^{(k)}$. The above analysis shows that:

$$\mathcal{W}_D^2(P_0, Q_0^{(l)}) \leq (1 - \alpha_0)^l \mathcal{W}_D^2(P_0, Q_0) + \sum_{j=0}^{l-1} (1 - \alpha_0)^j \mathcal{E}_1 \tag{10}$$

When $l$ is large enough, then $\mathcal{W}_D^2(P_0, Q_0^{(l)})$ converges to a value $\leq \frac{\mathcal{E}_1}{1-\alpha_0}$, which does not depend on the error added by the initial denoising steps. This is empirically demonstrated in Appendix L.3 for molecule generation, where ReDeNoise helps improve performance up to 6 rounds after which it stabilizes.

# D    Reduction to Binary Classification

The objective is to learn $\mathbb{P}\left(S_{i_t}^{(t)}|S_{-i_t}^{(t+1)} = \hat{s}_{-i_t}^{(t)}\right)$. Lemma 3.1 from Varma et al. (2025) can be used to simplify this objective:

**Lemma D.1.** *Let $s \in \mathcal{S}^L$. Then, for $x \in \mathcal{X}$ and discrete $s_{i_t}^{(t)}$, we can write $\mathbb{P}\left(S_{i_t}^{(t)} = x|S_{-i_t}^{(t+1)} = s_{-i_t}\right)$ as :*

$$\frac{\mathbb{P}(Z_t = x)}{\mathbb{P}(Z_t = \phi)}\left(\frac{1}{\mathbb{P}\left(Z_t = x|S_{-i_t}^{(t+1)} = s_{-i_t}, S_{i_t}^{(t+1)} = x\right)} - 1\right)$$

*where $\left(S^{(0)}, \dots S^{(T)}\right)$ denote the random vectors corresponding to the forward process and $Z_t$ is distributed according to the distribution $\Pi_t$.*

Hence, it is sufficient for the model to learn $\mathbb{P}\left(Z_t = x|S_{-i_t}^{(t+1)} = s_{-i_t}, S_{i_t}^{(t+1)} = x\right)$ for all $x \in \mathcal{X}$. This can be formulated as a binary classification task: Given $s_{-i_t}^{(t+1)}$ and $s_{i_t}^{(t+1)} = x$ as the input, predict whether $Z_t = x$ or $Z_t = \phi$. Hence, we minimize the binary cross-entropy loss: $\mathcal{L}_{BCE}\left(\theta; s_{-i_t}^{(t+1)}, t\right) = -\mathbf{1}_{z_t \neq \phi} \log\left(g_\theta^x\left(s_{-i_t}^{(t+1)}, t\right)\right) - \mathbf{1}_{z_t = \phi} \log\left(1 - g_\theta^x\left(s_{-i_t}^{(t+1)}, t\right)\right)$ where $g_\theta^x(\cdot)$ denotes the logit corresponding to token $x$.

# E   Learning Density through Score Functions

Suppose we are given a sample $s^{(t,k)}$ from the forward process at sequence time $t$ and element time $k$. Let $\stackrel{d}{=}$ denote equality in distribution. Consider the Orstein-Uhlenbeck Process:

$$dX_\tau = -X_\tau d\tau + \sqrt{2}dB_\tau$$

with standard Brownian motion $B_\tau$ and initial condition $x_0 = s_{i_t}^{(t)} \in \mathbb{R}^{d_{i_t}}$. Then $X_{\tau_0}|s^{(t,k)} \stackrel{d}{=} S_{i_t}^{(t,k+1)}|s^{(t,k)}$ whenever $\tau_0 = \frac{1}{2}\log\left(\frac{1}{1-\beta(t,k)}\right)$. Based on the observations in Song et al. (2021); Ho et al. (2020), the reverse SDE given by

$$X_\tau^{\mathsf{rev}} = X_\tau^{\mathsf{rev}}d\tau + 2\nabla \log q_{\tau_0-\tau}(X_\tau^{\mathsf{rev}}|s_{-i_t}^{(t)})\tau + \sqrt{2}dB_\tau \tag{11}$$

is such that if $x_0^{\mathsf{rev}} = s_{i_t}^{(t+1)}$ then $X_{\tau_0}^{\mathsf{rev}}|s^{(t+1)} \stackrel{d}{=} S_{i_t}^{(t)}|s^{(t+1)}$ where $q_\tau(\cdot|s^{(t+1)})$ is the conditional density function of $X_\tau$. We use DDPM (Ho et al., 2020) to sample from $\mathbb{P}(s_{i_t}^{(t,k)} = \cdot|s^{(t,k+1)})$ by learning the score function $\nabla \log q_\tau(\cdot|s_{-i_t}^{(t,k+1)})$ and then discretizing the reverse SDE in Equation equation 11. With the discretization we consider, the reverse process becomes:

$$\hat{s}_{i_t}^{(t,k)} = \frac{\left(\hat{s}_{i_t}^{(t,k+1)} - \beta(t,k+1)p(s^{(t,k+1)})\right)}{\sqrt{1-\beta(t,k+1)}} + \sqrt{\beta(t,k+1)}\epsilon'$$

where $\epsilon' \in \mathcal{N}(0,\mathbf{I})$ and $p(s^{(t,k+1)})$ is the score function $\nabla \log q(S_{i_t}^{(t,k+1)}|s_{-i_t}^{(t,k+1)})$.

# F    Connection between $\beta$ and $\bar{\alpha}$

From Section 3.1, we have:

$$s_{i_t}^{(t,k+1)} = \left(\sqrt{1 - \beta(t,k)}\right) s_{i_t}^{(t,k)} + \left(\sqrt{\beta(t,k)}\right)\epsilon \tag{12}$$

where $s_{i_t}^{(t,0)} = s_{i_t}^{(t)}$ and $s_{i_t}^{(t,K_{i_t}^t)} = s_{i_t}^{(t+1)}$. Define $\alpha(t,k) = 1 - \beta(t,k)$. Then, we have:

$$s_{i_t}^{(t,k+1)} = \left(\sqrt{\prod_{k'=0}^{k} \alpha(t,k')}\right) s_{i_t}^{(t,0)} + \left(\sqrt{1 - \prod_{k'=0}^{k} \alpha(t,k')}\right)\epsilon$$

Recall from Section 4.2 that:

$$s_{i_t}^{(t,k+1)} = \left(\sqrt{\bar{\alpha}(t,k)}\right) s_{i_t}^{(0)} + \left(\sqrt{1 - \bar{\alpha}(t,k)}\right)\epsilon \tag{13}$$

Rewriting equation 12 by merging Gaussians, we have:

$$s_{i_t}^{(t,k+1)} = \left(\sqrt{\prod_{t'=0}^{t-1}\prod_{k'=0}^{K_i^{t'}} \alpha(t',k') \prod_{k''=0}^{k} \alpha(t,k'')}\right) s_{i_t}^{(0,0)} + \left(\sqrt{1 - \prod_{t'=0}^{t-1}\prod_{k'=0}^{K_i^{t'}} \alpha(t',k') \prod_{k''=0}^{k} \alpha(t,k'')}\right)\epsilon$$

Comparing with equation 13, we have:

$$\bar{\alpha}(t,k) = \prod_{t'=0}^{t-1}\prod_{k'=0}^{K_i^{t'}} \alpha(t',k') \prod_{k''=0}^{k} \alpha(t,k'') \tag{14}$$

## G   Forward process: Generating $s^{(t)}$ directly

For training the model for denoising at sequence time $t$ (and element time $k$ if we are denoising a continuous vector), we need access to:

- $(s^{(t)}, s^{(t+1)})$ if $s^{(t)}_{i_t}$ is discrete

- $(s^{(t,k+1)}, \epsilon)$ if $s^{(t,k)}_{i_t}$ is continuous

Note that $\epsilon$ is as defined in equation 2. Once you have $s^{(t)}$, $s^{(t+1)}$ can be generated by applying one discrete noising step, by sampling $z_t$. $\epsilon$ can be directly computed using $s^{(t,k+1)}_{i_t}$ and $s^{(0)}_{i_t}$. Hence, if we can directly sample $s^{(t,k)}$ without going through all intermediate timesteps, the model can be trained efficientlys.

Let us denote by $m^{(t,k)}_j$ the total number of times an element at position $j$ has been noised by sequence time $t$ and element time $k$. Further, let $\tau^t_j = \{t' \in \{0, 1, \ldots, t\}; i_{t'} = j\}$ denote the set of all sequence timesteps $t'$ at which position $j$ was visited prior to (and including) sequence time $t$.

For discrete elements, $m^{(t,k)}_j = |\tau^t_j|$. That is, for discrete elements, the number of noising steps is equal to the number of visits at that position by time $t$ (Note that element time $k$ is irrelevant for discrete noising ).

For continuous vectors, $m^{(t,k)}_j = \sum_{t' \in \tau^t_j} K^{t'}_j$, provided $j \neq i_t$. That is, for continuous vectors which are not being noised at time $t$, the total number of noising steps are obtained by summing up the element times for all prior visits at that position. For $j = i_t$, $m^{(t,k)}_j = \sum_{t' \in \tau^t_j - \{t\}} K^{t'}_j + k$. That is, if a continuous vector is being noised at time $t$, the number of noising steps for that vector is obtained by summing up the element times for all prior visits as well as the current element time.

Since the forward process for each element is independent of other elements, to describe the generation of $s^{(t)}$ (or $s^{(t,k)}$) from $s^0$, it is sufficient to describe generating $s^{(t)}_j$ (or $s^{(t,k)}_j$) from $s^{(0)}_j$ individually for each $j$.

**If $s^{(t)}_j$ is discrete:**   Recall from Section 3.1 that $\Pi_t(\phi)$ denotes the probability of sampling the token $\phi$ at sequence time $t$. Further, assume $\Pi_t(\cdot|\mathcal{X})$ is same for all $t$. Define $p^t_j = 1 - \prod_{t' \in \tau^t_j}(1 - \Pi_{t'}(\phi))$. Sample $z \sim \Pi(\cdot|\mathcal{X})$ Then:

$$s^{(t)}_j = \begin{cases} s^{(0)}_j, & \text{with probability } 1 - p^t_j \\ z, & \text{with probability } p^t_j \end{cases}$$

The above follows from the fact that each flip for $t' \in \tau^t_j$ is an independent Bernoulli trial and hence, even if there is one success among these $m^{(t,k)}_j$ trials, the token is flipped according to $\Pi(\cdot|\mathcal{X})$.

**If $s^{(t,k)}_j$ is continuous:**   Following Section 4.2, we define the continuous noise schedule for the continuous vector at position $j$ as $\beta_j = \{\beta_i; i \in \{0, 1, \ldots, m^{(T,0)}_j - 1\}\}$. Let $\beta_j[i]$ denote the $i^{\text{th}}$ element of $\beta_j$. Define $\bar{\alpha}_j = \{\prod_{i' \leq i}(1 - \beta_j[i']); i \in \{0, 1, \ldots, m^{(T,0)}_j - 1\}\}$. Let $\bar{\alpha}_j[i]$ denote the $i^{\text{th}}$ element of $\bar{\alpha}_j$. Then, we have:

$$s^{(t,k)}_{i_t} = \left(\sqrt{\bar{\alpha}_{i_t}[m^{(t,k)}_{i_t}]}\right) s^{(0)}_{i_t} + \left(\sqrt{1 - \bar{\alpha}_{i_t}[m^{(t,k)}_{i_t}]}\right) \epsilon$$

where $\epsilon \sim \mathcal{N}(0, \mathbf{I})$.

The forward process can thus be thought of as the block `FwdPrcs` with the following input and output:

Input: $(s^{(0)}, t, \{i_\tau\}^t_{\tau=0}, \{\Pi_\tau\}^t_{\tau=0})$     Output: $(s^{(t)}, z_t, s^{(t+1)})$   if $s^{(0)}_{i_t}$ is discrete

Input: $(s^{(0)}, t, k, \{i_\tau\}^t_{\tau=0}, \{\beta_j\}^{L_2}_{j=L_1+1})$     Output: $(s^{(t,k+1)}, \epsilon)$   if $s^{(0)}_{i_t}$ is continuous

## H    Model Training and Inference: Pseudocode

We use the binary classification based loss for describing the training of the model to do discrete denoising since this leads to better results. Note that for this, from , the input to the model should be $s_{-i_t}^{(t+1)}$ and the model should predict $\mathbb{P}\left(Z_t = x | S_{-i_t}^{(t+1)} = s_{-i_t}, s_{i_t}^{(t+1)} = x\right)$ for all $x \in \mathcal{X}$. To do this efficiently, we adapt the masking strategy from Varma et al. (2025). Define a token $\omega \notin \mathcal{X}$. Let $\tilde{\mathcal{X}} = \mathcal{X} \cup \omega$. Let $\tilde{s}^{(t+1)} \in \tilde{\mathcal{S}}_L$, where $\tilde{\mathcal{S}}_L = \tilde{\mathcal{X}}^{L_1} \times_{i=L_1+1}^{L} \mathbb{R}^{d_i}$, be defined as: $\tilde{s}_{-i_t}^{t+1} = s_{-i_t}^{t+1}$ and $\tilde{s}_{i_t}^{t+1} = \omega$. The neural network $f_\theta$ then takes as input: time tuple $(t, k)$, noising position $i_t$, sequence $\tilde{s}^{t+1}$ (or $\tilde{s}^{t,k+1}$ if $i_t$ corresponds to a continuous vector). The time tuple $(t, k)$ is $(t, 0)$ if the element under consideration is discrete since discrete tokens only have one noising step. The model has $|\mathcal{X}|$ logits corresponding to *each* discrete token (and hence a total of $L_1|\mathcal{X}|$ logits) and $\mathbb{R}^{d_i}$ dimensional vectors corresponding to *each* continuous vector (and hence a total of $L_2$ continuous vectors). $i_t$ is necessary for the model to decide which output needs to be sliced out: we use $f_\theta^{i_t}$ to denote the output of the model corresponding to the element at position $i_t$ (which could either be discrete or continuous). Further, we use $f_\theta^{(i_t, s_{i_t}^{(t+1)})}$ to denote the logit corresponding to position $i_t$ and token $s_{i_t}^{t+1}$, provided $i_t$ corresponds to a discrete token.

We can then write the pseudocode for training as follows:

---
**Algorithm 2** Model Training
---
**Input:** Dataset $\mathcal{D}$, model $f_\theta$ , forward process block `FwdPrcs`, optimizer `opt`, total sequence timesteps $T$, noise positions $\{i_t\}_{t=0}^{T-1}$, discrete noise schedule $\{\Pi_t\}_{t=0}^{T-1}$, continuous noise schedule $\{\beta_j\}_{j=L_1+1}^{L_2}$, continuous noising steps $\{K_j^t\}_{j=L_1+1,t=0}^{j=L_2,t=T-1}$

**Output:** trained model parameters $\theta$

1: **for** each iteration: **do**
2:    sample $s^{(0)}$ from $\mathcal{D}$
3:    sample $t$ from $[0, 1, \ldots, T-1]$
4:    **if** $\hat{s}_{i_t}^{(0)}$ is discrete **then**
5:       $(s^{(t)}, z_t, s^{(t+1)}) = \texttt{FwdPrcs}(s^{(0)}, t, \{i_\tau\}_{\tau=0}^t, \{\Pi_\tau\}_{\tau=0}^t)$
6:       construct $\tilde{s}^{t+1}$ from $s^{t+1}$
7:       compute the BCE loss:

$$\mathcal{L} = -\mathbf{1}_{z_t \neq \phi} \log\left(f_\theta^{(i_t, s_{i_t}^{(t+1)})}\left(\tilde{s}^{(t+1)}, t, i_t\right)\right) - \mathbf{1}_{z_t = \phi} \log\left(1 - f_\theta^{(i_t, s_{i_t}^{(t+1)})}\left(\tilde{s}^{(t+1)}, t, 0, i_t\right)\right)$$

8:    **else**
9:       sample $k$ from $[0, 1, \ldots, K_{i_t}^t - 1]$
10:       $(s^{(t,k+1)}, \epsilon) = \texttt{FwdPrcs}(s^{(0)}, t, k, \{i_\tau\}_{\tau=0}^t, \{\beta_j\}_{j=L_1+1}^{L_2})$
11:       compute the MSE loss:
$$\mathcal{L} = \left\| \epsilon - f_\theta^{i_t}\left(s^{(t,k+1)}, t, k, i_t\right) \right\|_2^2$$

12:    **end if**
13:    $\theta \leftarrow \texttt{opt.update}(\theta, \nabla_\theta \mathcal{L})$
14: **end for**
---

Recall that $\hat{s}^{(t)}$ represents the sequence from the reverse process at time $t$ and $P_T = \Pi \left( \cdot | \mathcal{X} \right)^{L_1} \times_{i=L_1+1}^{L} \mathcal{N} \left( 0, \mathbf{I}_{d_i} \right)$ denotes the stationary distribution of the forward process. If the training of the model is perfect, we will have $\hat{s}^{(0)} \sim \pi$. Then the pseudocode for inference:

---

**Input:** total sequence timesteps $T$, noise positions $\{i_t\}_{t=0}^{T-1}$, discrete noise schedule $\{\Pi_t\}_{t=0}^{T-1}$, continuous noise schedule $\{\beta_j\}_{j=L_1+1}^{L_2}$, continuous noising steps $\{K_j^t\}_{j=L_1+1,t=0}^{j=L_2,t=T-1}$

**Output:** $\hat{s}^{(0)}$

1: sample $\hat{s}^{(T)} \sim P_T$
2: **for** $t$ in $[T-1, T-2, \cdots, 0]$ **do**
3:     **if** $\hat{s}_{i_t}^{(t+1)}$ is discrete **then**
4:         construct $\tilde{s}^{(t+1)}$ from $\hat{s}^{(t+1)}$
5:         get $\hat{y} = f_\theta^{i_t} \left( \tilde{s}^{(t+1)}, t, i_t \right)$ { $\hat{y}$ denotes the vector of $|\mathcal{X}|$ logits corresponding to position $i_t$}
6:         compute $\hat{\mathbb{P}} \left( s_{i_t}^{(t)} = a | s_{-i_t}^{(t+1)} \right) = \frac{\Pi_t(a)}{\Pi_t(\phi)} \left( \frac{1}{\hat{y}^{(a)}} - 1 \right)$ for all $a \in \mathcal{X}$ { $\hat{y}^{(a)}$ denotes logit corresponding to token $a$}
7:         sample $\hat{s}_{i_t}^{(t)} \sim \hat{\mathbb{P}} \left( s_{i_t}^{(t)} = a | s_{-i_t}^{(t+1)} \right)$
8:         set $\hat{s}_{-i_t}^{(t)} = \hat{s}_{-i_t}^{(t+1)}$
9:     **else**
10:        set $\hat{s}^{(t,K_{i_t}^t)} = \hat{s}^{(t+1)}$
11:        **for** $k$ in $[K_{i_t}^t - 1, K_{i_t}^t - 2, \cdots, 0]$ **do**
12:            get $\epsilon_\theta = f_\theta^{i_t} \left( \hat{s}^{(t,k+1)}, t, k, i_t \right)$ { $f_\theta^{i_t}$ denotes the continuous vector corresponding to position $i_t$}
13:            **if** t = k = 0 **then**
14:                get $\epsilon = 0$
15:            **else**
16:                get $\epsilon \sim \mathcal{N}(0, \mathbf{I})$
17:            **end if**
18:            set $\hat{s}_{i_t}^{(t,k)} = \frac{\left( \hat{s}_{i_t}^{(t,k+1)} - \beta_{i_t}(t,k+1)\epsilon_\theta \right)}{\sqrt{1-\beta_{i_t}(t,k+1)}} + \left( \sqrt{\beta_{i_t}(t,k+1)} \right) \epsilon$
19:            set $\hat{s}_{-i_t}^{(t,k)} = \hat{s}_{-i_t}^{(t,k+1)}$
20:        **end for**
21:        set $\hat{s}^{(t)} = \hat{s}^{(t,0)}$
22:     **end if**
23: **end for**

---

# I Model Architecture

Our proposed architecture, which we refer to as Discrete-Continuous (Dis-Co) DiT, is illustrated in Figure 2.

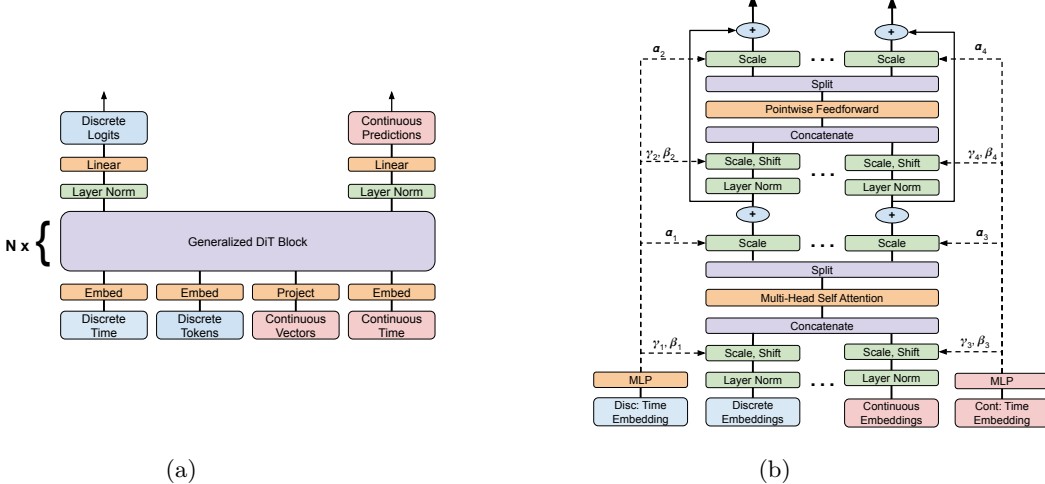

(a)  (b)

Figure 2: **Dis-Co DiT Architecture:** (a) illustrates overall architecture, with both discrete and continuous inputs and outputs (b) shows detailed architecture of a single block, where time information is incorporated through adaptive layer normalization.

Figure 2a gives a high level overview of the proposed Dis-Co DiT architecture. Just like DiT, we feed in the discrete tokens and corresponding discrete time as input to the Dis-Co DiT block; however, we now also feed in the continuous vector inputs and corresponding continuous time. Also note that time is now the tuple $(t, k)$, where $t$ is the sequence time and $k$ is the element time. For discrete elements $k = 0$ always. Time is now embedded through an embedding layer similar to DiT; discrete tokens are also embedded through an embedding layer. Continuous vectors are projected using a linear layer into the same space as the discrete embeddings; these projected vectors are referred to as continuous embeddings. Discrete embeddings, continuous embeddings and their corresponding time embeddings are then passed into the Dis-Co DiT blocks. Following DiT, the outputs from the Dis-Co DiT blocks are then processed using adaptive layer normalization and a linear layer to obtain the discrete logits and continuous predictions.

Figure 2b details the structure of a single Dis-Co DiT block. The discrete and continuous time embeddings are processed by an MLP and are used for adaptive layer normalization, adaLN-Zero, following DiT. The discrete and continuous embedding vectors, after appropriate adaptive layer normalization, are concatenated and passed to the Multi-Head Self Attention Block. The output from the Self Attention block is again split into discrete and continuous parts, and the process is then repeated with a Pointwise Feedforward network instead of Self Attention. This output is then added with the output from Self Attention (after scaling) to get the final output from the Dis-Co DiT block.

**Generating Time Embeddings:** Assume you are embedding the time tuple $(t, k)$ ($k = 0$ for discrete). Following DiT, we compute the vector $d$ whose $i^{th}$ element is given by:

$$d[i] = k * f^{\frac{-i}{d_{in}-1}}$$

where $k$ is the element time, $f$ is the frequency parameter (set to 10000 in all our experiments) and $d_{in}$ is the time embedding input dimension (set to 256 in all our experiments). Similarly, we compute the vector $c$ whose $i^{th}$ element is given by:

$$c[i] = t * (T_C f)^{\frac{-i}{d_{in}-1}}$$

where $t$ is the sequence time, $T_C$ is a frequency multiplier designed to account for the fact that multiple continuous noising steps happen for a single discrete flip. In our experiments, we set $T_C = K_{i_t}^t$. Once we

have these vectors, we construct the following vector:

$$y = [\sin(d) \ \cos(d) \ \sin(c) \ \cos(c) \ ]$$

i.e., we concatenate the vectors after applying sin and cos elementwise. This vector $y$ is then passed through 2 MLP layers to get the final time embedding.

## J Boolean Satisfiability Problem

### J.0.1 Background

The Boolean Satisfiability (SAT) problem is the task of determining whether there exists a binary assignment to the variables of a given Boolean expression (in Conjunctive Normal Form (CNF)) that makes it evaluate to *True*. SAT is a canonical NP-Complete problem (Cook, 1971) and underlies a broad range of real-world applications in formal hardware/software verification, resource scheduling, and other constraint satisfaction tasks (Clarke et al., 2001; Gomes et al., 2008; Vizel et al., 2015).

Our goal is to find a valid assignment for the Boolean variables, when the given CNF formula is satisfiable. Let $n$ be the number of variables and $m$ the number of clauses. In Random $k$-SAT, a well-studied variation of SAT, the relative difficulty of an instance is determined by the clause density $\frac{m}{n}$. There is a sharp transition between satisfiable and unsatisfiable instances of random 3-SAT at the critical clause density $\alpha_{\text{sat}}(k=3)$, when m is set close to $m = 4.258n + 58.26n^{-\frac{2}{3}}$ (Ding et al., 2015). Following the setup of Ye et al. (2025), we choose $m$ close to this threshold to focus on relatively hard random 3-SAT instances.

### J.0.2 Experimental Setup

**Datasets:** We consider two experimental setups:

Setup 1 (Single $n$): We follow the train and test partitions from Ye et al. (2025), which provide separate datasets for $n = 5, 7$, and 9, for direct comparison. Specifically, $n = 5$ and $n = 7$, use a training set of 50K samples each, while for $n = 9$, the training set consists of 100K samples.

Setup 2 (Combined $n$): We then move to a more challenging, large-scale setting by extending the range of $n$ up to 20. Following the same generation procedure, for each $n$ in $6, 7, \ldots, 20$, we generate 1M training samples, resulting in a combined dataset of 15M instances. In this setup, we train a single model on the aggregated data covering all $n$ from 6 to 20. Figure 3 illustrates how the model's accuracy varies with $n$ under different model sizes.

More details on data generation and model configuration are provided in Appendix J.1.

**Baselines:** We compare against two types of baseline models: 1) Autoregressive Models with a GPT-2 architecture (Radford et al., 2019) trained from scratch and 2) Discrete diffusion models in Ye et al. (2025) (MGDM) that applies adaptive sequence- and token-level reweighting to emphasize difficult subgoals in planning and reasoning. MGDM has demonstrated strong performance on tasks such as Sudoku and Boolean Satisfiability compared to standard autoregressive and discrete diffusion approaches.

Table 4: **SAT:** Accuracy with increasing number of variables $n$. Separate model trained for each $n$

| Method | Params | $n = 5$ | $n = 7$ | $n = 9$ |
|---|---|---|---|---|
| GPT-2 Scratch | 6M | 97.6 | 85.6 | 73.3 |
| MGDM | 6M | 100.0 | 95.9 | 87.0 |
| GGM | 6M | 100.0 | 98.0 | 94.5 |
| | 85M | - | 99.9 | 99.9 |

### J.0.3 Results

In Table 4, we see that our method consistently outperforms the autoregressive (GPT-2) and diffusion-based (MGDM) baselines across different choices for $n$. This performance gap is more pronounced for larger $n$: at $n = 9$, our model achieves 94.5% accuracy, compared to 87.0% for MGDM and 73.3% for GPT-2. Scaling the model to 85M parameters further reaches near-perfect accuracy (99.9%) for $n = 7$ and $n = 9$, thus highlighting the crucial role of model capacity in handling complex SAT instances.

For *Setup 2*, Figure 3 reveals a steep accuracy drop for the 6M-parameter model; it starts declining around $n = 8$ and approaches zero by $n = 12$. In contrast, the 85M-parameter model remains robust up to $n = 18$,

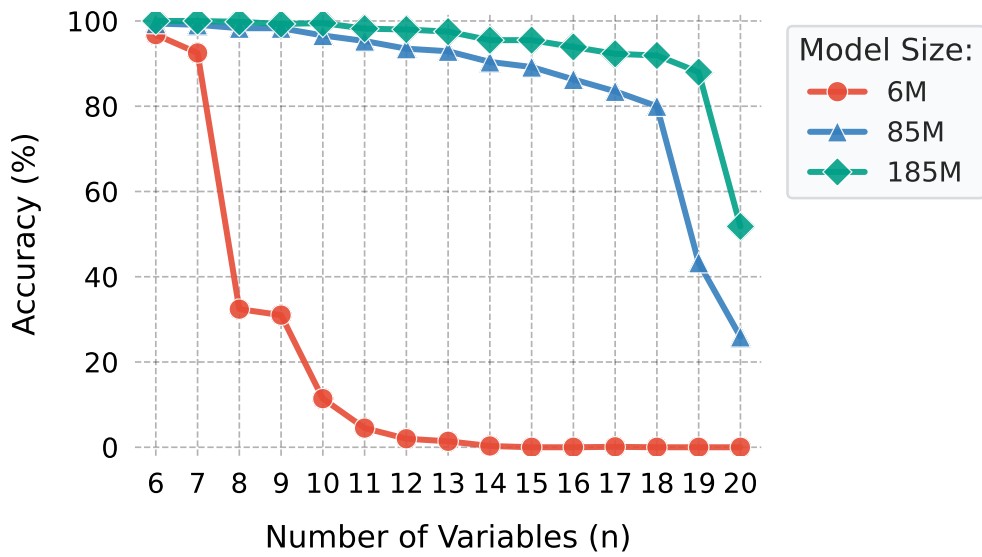

Figure 3: **SAT:** Accuracy for different number of variables across model sizes trained on a combined dataset for n in $6, 7 \ldots, 20$.

and an even larger 185M-parameter model sustains high accuracy near $n = 19$. This degradation trend aligns with the theoretical hardness of random 3-SAT, where solution spaces become exponentially sparse as $n$ increases. Larger models postpone this accuracy drop underscoring a direct relationship between parameter count and combinatorial reasoning capacity.

### J.1 Training Details

We trained models of three different sizes (6M, 85M, and 185M parameters), whose configurations are summarized in Table 5. Each model was trained for 1M steps on the combined dataset with $n \in 6, \ldots, 20$. For the experiments where a separate model was trained for each $n$ (corresponding to Table 1), the batch size was increased from 8192 to 16384 and trained for 200K steps. A gradual noising schedule of $[0.99, 0.9, 0.8, 0.5, 0.5, 0.25]$ was used for the discrete noising process in all SAT experiments.

| Parameter | 6M | 85M | 185M |
|---|---|---|---|
| Number of DiT Blocks | 4 | 12 | 24 |
| Number of Heads | 8 | 12 | 16 |
| Model Dimension | 336 | 744 | 768 |
| MLP Dimension | 1344 | 2976 | 3072 |
| Time Embedding Input Dim | 256 | 256 | 256 |
| Time Embedding Output Dim | 128 | 128 | 128 |
| Learning Rate | 2e-4 | 7.5e-5 | 5e-5 |
| Batch Size | 8192 | 8192 | 4096 |

Table 5: Model Configurations for Different Parameter Sizes for Boolean Satisfiability Problem

Here DiT Block (Peebles & Xie, 2023) is a modified transformer block designed to process conditional inputs in diffusion models. For Boolean Satisfiability (SAT), these blocks evolve variable assignments and clause states while incorporating diffusion timestep information through specialized conditioning mechanisms.

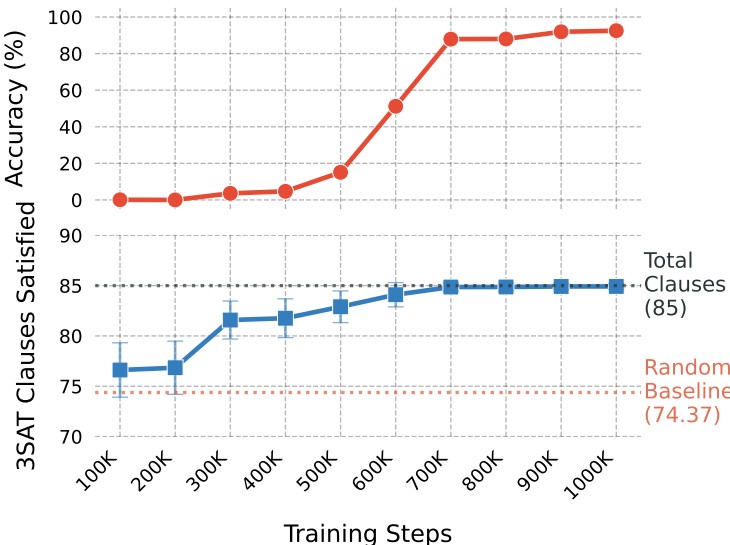

Figure 4: Evolution of the model's SAT accuracy and number of satisfied clauses over training for random 3-SAT instances with $n = 18$ on 185M model.

Adaptive Layer Norm (adaLN-Zero) (Xu et al., 2019): Dynamically adjusts normalization parameters using timestep embeddings:

$$\text{AdaLN}(h, t) = t_s \cdot \text{LayerNorm}(h) + t_b \tag{15}$$

where $t_s$, $t_b$ are learned projections from timestep $t$. The *adaLN-Zero* variant initializes residual weights ($\alpha$) to zero, preserving identity initialization for stable training.

Time-conditioned MLP: Processes normalized features with gated linear units (GLU), scaled by the diffusion timestep.

We use the AdamW optimizer (Loshchilov & Hutter, 2019) (with $\beta_1 = 0.9$, $\beta_2 = 0.999$ and $\epsilon = 10^{-8}$) with no weight decay and with no dropout. We use EMA with decay 0.9999.

### J.2 Data Generation

We follow the procedure of Ye et al. (2025) to create a large dataset of 15M satisfiable 3-SAT instances covering $n \in 6, \ldots, 20$. Each instance is generated by:

1. Sampling clauses where each clause has exactly three variables, chosen uniformly at random from the $n$ available.

2. Randomly deciding whether each variable in the clause appears in complemented or non-complemented form.

After generating the clauses, we run a standard SAT solver to ensure each instance is satisfiable, discarding any unsatisfiable cases. Finally, the data is split into training and test sets, with multiple checks to prevent overlap.

### J.3 Accuracy Trend During Training

Figure 4 illustrates how the SAT accuracy evolves over training for a model trained on instances, showing for $n = 18$ as a representative example. In the early stages (roughly the first half of training), the accuracy remains near zero, even as the model steadily improves in satisfying individual clauses. This indicates that

the model initially learns partial solutions that satisfy a growing fraction of the clauses. Once the model begins consistently satisfying nearly all clauses in an instance, accuracy jumps sharply, reflecting that the assignments finally meet all the constraints simultaneously.

### J.4  Compute Requirements:

All training was conducted on TPUv6e pods[Cloud], with each pod consisting of 8 TPU chips. For experiments on individual datasets (n=5, 7, 9), models were trained on a single pod, with training times ranging upto 2 days, the longest being the 85M parameter model on the n=9 dataset. For the combined dataset (n=6 to 20), the largest 185M parameter model utilized two pods and training took approximately 10 days. More information about the compute architecture and configuration at [link].

# K    Layout Generation

## K.1    Additional Results

Table 6: **Layout Generation:** Additional metrics on the RICO and PubLayNet datasets.

| | RICO | | | | | | | PubLayNet | | | | | |
|---|---|---|---|---|---|---|---|---|---|---|---|---|---|
| | Unconditioned | | Category Conditioned | | Category+Size Conditioned | | | Unconditioned | | Category Conditioned | | Category+Size Conditioned | |
| Method | Ali→ | Ove→ | Ali→ | Ove→ | Ali→ | Ove→ | Method | Ali→ | Ove→ | Ali→ | Ove→ | Ali→ | Ove→ |
| LayoutTransformer | 0.037 | 0.542 | - | - | - | - | LayoutTransformer | 0.067 | 0.005 | - | - | - | - |
| LayoutFormer**++** | 0.051 | 0.546 | **0.124** | 0.537 | - | - | LayoutFormer**++** | 0.228 | 0.001 | **0.025** | 0.009 | - | - |
| NDN-none | - | - | 0.560 | 0.550 | - | - | NDN-none | - | - | 0.350 | 0.170 | - | - |
| LayoutDM | 0.143 | 0.584 | 0.222 | 0.598 | 0.175 | 0.606 | LayoutDM | 0.180 | 0.132 | 0.267 | 0.139 | 0.246 | 0.160 |
| DLT | 0.271 | 0.571 | 0.303 | 0.616 | 0.332 | 0.609 | DLT | 0.117 | 0.036 | 0.097 | 0.040 | 0.130 | 0.053 |
| LayoutDiffusion | 0.069 | 0.502 | **0.124** | 0.491 | - | - | LayoutDiffusion | 0.065 | **0.003** | 0.029 | **0.005** | - | - |
| LayoutFlow | **0.150** | 0.498 | 0.176 | 0.517 | 0.283 | 0.523 | LayoutFlow | **0.057** | 0.009 | 0.037 | 0.011 | **0.041** | 0.031 |
| Ours | 0.198 | **0.443** | 0.215 | **0.461** | **0.204** | **0.490** | Ours | 0.094 | 0.008 | 0.088 | 0.013 | 0.081 | **0.027** |
| | Alignment | | | Overlap | | | | Alignment | | | Overlap | | |
| Validation Data | 0.093 | | | 0.466 | | | Validation Data | 0.022 | | | 0.003 | | |

Alignment and Overlap capture the geometric aspects of the generations. As per Guerreiro et al. (2025), we judge both metrics with respect to a reference dataset, which in our case is the validation dataset. While, our model seems to do well with respect to Overlap on the RICO dataset, in general we see that there is no consistent "winner" with respect to these metrics among models. Further, note that most of the reported models use specialized losses to ensure better performance specifically with respect to these metrics; our model achieves comparable performance despite not using any specialized losses. Our framework can be used in tandem with domain-specific losses to improve the performance on these geometric metrics.

## K.2    Generated Examples

Table 7 shows generated samples on PubLayNet dataset on the three tasks of Unconditioned, Category-conditioned and Category+Size conditioned.

**Unconditioned Generation**    **Category-conditioned Generation**    **Category+Size-conditioned Generation**

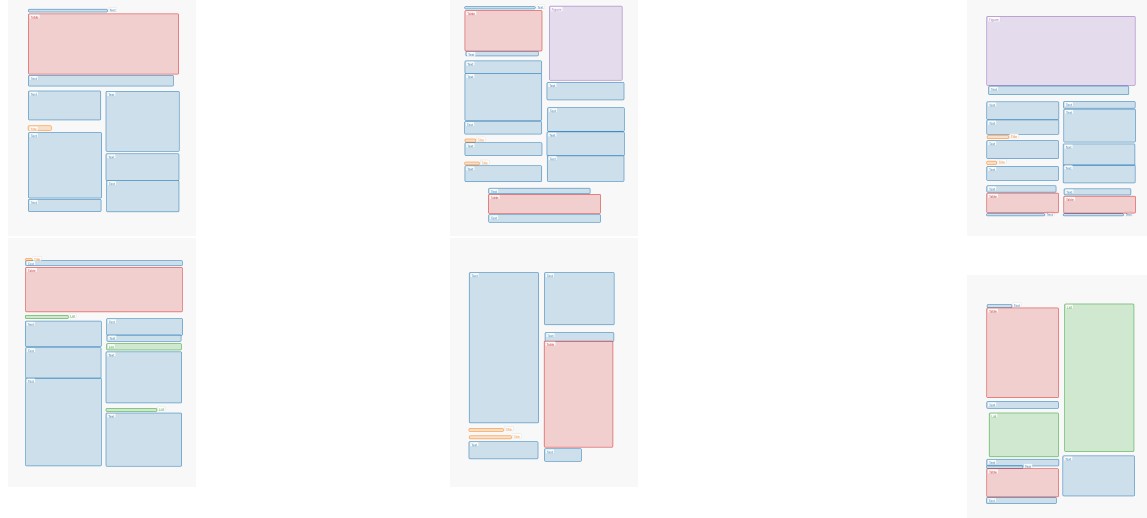

Table 7: Generated Layouts on PubLayNet Dataset

## K.3    Training Details

We train a Dis-Co DiT model with the configuration in Table 8.

| Number of Generalized DiT Blocks | 6 |
|---|---|
| Number of Heads | 8 |
| Model Dimension | 512 |
| MLP Dimension | 2048 |
| Time Embedding Input Dimension | 256 |
| Time Embedding Output Dimension | 128 |

Table 8: Model configuration for Layout Generation

We use the AdamW optimizer (Loshchilov & Hutter, 2019) (with $\beta_1 = 0.9$, $\beta_2 = 0.999$ and $\epsilon = 10^{-8}$) with no weight decay and with no dropout. We use EMA with decay 0.9999. We set the initial learning rate to 0 and warm it up linearly for 8000 iterations to a peak learning rate of $10^{-4}$; a cosine decay schedule is then applied to decay it to $10^{-6}$ over the training steps. For PubLayNet, we train for 4 Million iterations with a batch size of 4096, whereas for RICO, we train for 1.1 Million iterations with a batch size of 4096. By default, the sequence is noised for 4 rounds ($T = 160$); each continuous vector is noised 200 times per round. We use pad tokens to pad the number of elements to 20 if a layout has fewer elements.

**Data sampling and pre-processing:**
Since we train a single model for all three tasks (unconditional, class conditioned, class and size conditioned), we randomly sample layouts for each task by applying the appropriate binary mask required for the state-space doubling strategy. We begin training by equally sampling for all three tasks; during later stages of training, it may help to increase the fraction of samples for harder tasks to speed up training. For instance, we found that for the RICO dataset, doubling the fraction of samples for unconditional generation after 700k iterations results in better performance in unconditional generation (while maintaining good performance in the other two tasks) when training for 1.1 Million iterations. Further, each bounding box is described as $[x_i, y_i, l_i, w_i]$, where $(x_i, y_i)$ denotes the positions of the upper-left corner of the bounding box and $(l_i, w_i)$ denotes the length and width of the bounding box respectively. Note that $0 \le x_i, y_i, l_i, w_i \le 1$ since the dataset is normalized. We further re-parameterize these quantities using the following transformation:

$$g(x) = \log\left(\frac{x}{1-x}\right)$$

Note that we clip $x$ to $[10^{-5}, 1 - 10^{-5}]$ so that $g(x)$ is defined throughout. We then use this re-parameterized version as the dataset to train the diffusion model. While inference, the predicted vectors are transformed back using the inverse transformation:

$$h(x) = g^{-1}(x) = \left(\frac{e^x}{1 + e^x}\right)$$

## K.4   Ablations

Unless specified otherwise, all the results reported in ablations use top-$p$ sampling with $p = 0.99$ and do not use the ReDeNoise algorithm at inference. From preliminary experiments, we found top-p sampling and ReDeNoise to only have marginal effects on the FID score; hence, we did not tune this further. For all layout generation experiments, we noise the sequence in a round-robin fashion, and in each round, $\Pi(\phi)$ is constant for discrete tokens across all positions. Similarly, $K_{i_t}^t$ which is the number of continuous noising steps per round, is constant across all positions per round. Hence, from here on, we use sequences of length $r$. where $r$ is the total number of noising rounds to denote $\Pi(\phi)$ and $K_{i_t}^t$ values for that particular round. By default, we choose $\Pi(\phi)$ to be $[0.5, 0.5, 0.5, 0.5]$, where the 4 element sequence, which we refer to as the discrete noise schedule, denotes noising for 4 rounds with $\Pi(\phi)$ for the round chosen from the sequence. Similarly, the default value of $K_{i_t}^t$ is chosen to be $[200, 200, 200, 200]$, and we refer to this sequence as the continuous noising steps. Let us denote $\sum_t K_{i_t}^t$ as $K$. Note that $K$ is same across positions since we assume same number of continuous noising steps across positions per round. Given $K$, we define the following as the cosine schedule

for $\beta$ (denoted by $\mathrm{cosine}(a, b)$):

$$\beta(j) = b + 0.5(a - b)(1 + \cos\left(\left(\frac{j}{K}\right)\pi\right))$$

where $j$ is the total number of continuous noising steps at sequence time $t$ and element time $k$. We use $\mathrm{cosine}(0.0001, 0.03)$ as the default schedule. We also define a linear noise schedule for $\beta$ ($\beta$ (denoted by $\mathrm{lin}(a, b)$)):

$$\beta(j) = a + (b - a)(1 + (\left(\frac{j}{K}\right)))$$

Further, we report only the unconditional FID for PubLayNet/RICO in the ablations as this is the most general setting.

**Interleaving pattern/ Noise order:** We broadly consider two interleaving patterns. In the first pattern $I_1$, the bounding box vectors of each item is treated as a separate vector to form the interleaving pattern $[t_1, p_1, t_2, p_2, \ldots, t_n, p_n]$, where $t_i \in \mathbb{N}$ is the discrete item type and $p_i \in \mathbb{R}^4$ is its corresponding bounding box description ($p_i = [x_i, y_i, l_i, w_i]^\top$). This interleaving pattern leads to 20 discrete elements and 20 continuous vectors per layout, resulting in a sequence of length 40. In the second pattern $I_2$, the bounding box vectors of all the $n$ items were bunched together as a single vector to form the interleaving pattern $[t_1, t_2, \ldots, t_n, p^c]$, where $p^c \in \mathbb{R}^{4n}$ is a single vector which is formed by concatenating the bounding box vectors of all $n$ items. This interleaving pattern leads to 20 discrete elements and 1 continuous vector per layout, resulting in a sequence of length 21. We compare FID scores on unconditional generation on PubLayNet with these two interleaving patterns in Table 9.

| Interleaving Pattern | Disc. Noise Schedule | Cont. Noise Schedule | Cont. Noise Steps | FID |
|---|---|---|---|---|
| $I_1$ | $[0.5, 0.5, 0.5, 0.5]$ | $\mathrm{cosine}(0.0001, 0.03)$ | $[200, 200, 200, 200]$ | 8.76 |
| $I_2$ | $[0.5, 0.5, 0.5, 0.5]$ | $\mathrm{cosine}(0.0001, 0.03)$ | $[200, 200, 200, 200]$ | 14.21 |
| $I_2$ | $[0.35, 0.5, 0.5, 0.5]$ | $\mathrm{cosine}(0.0001, 0.03)$ | $[200, 200, 200, 200]$ | 13.59 |
| $I_2$ | $[0.75, 0.5, 0.5, 0.5]$ | $\mathrm{cosine}(0.0001, 0.03)$ | $[200, 200, 200, 200]$ | 13.99 |
| $I_2$ | $[0.99, 0.9, 0.8, 0.5, 0.5, 0.5]$ | $\mathrm{cosine}(0.0001, 0.03)$ | $[150, 150, 150, 150, 150, 150]$ | 25.38 |
| $I_2$ | $[0.9, 0.75, 0.5, 0.5, 0.25]$ | $\mathrm{cosine}(0.0001, 0.015)$ | $[500, 500, 500, 500, 500]$ | 17.86 |

Table 9: Ablation on Interleaving Pattern

We see that despite tuning multiple hyperparameters for noise schedules, $I_2$ leads to worse results than having $I_1$. Hence, we use the interleaving pattern $I_1$ separate for all further experiments.

**$|\mathcal{X}|$-ary classification v/s Binary classification:** We compare the two strategies for training the discrete denoiser, $|\mathcal{X}|$-ary classification and Binary classification (as described in 4), on the unconditional generation task in the RICO dataset. The results are given in Table 10.

| Discrete Loss Considered | FID |
|---|---|
| $|\mathcal{X}|$-ary Cross Entropy | 3.51 |
| Binary Cross Entropy | 2.62 |

Table 10: Ablation on choice of discrete loss function

**Discrete and continuous noise schedules:** We evaluate the unconditional FID scores on PubLayNet and RICO for multiple configurations of discrete and continuous noise schedules. We report the results in Tables 11 and 12.

From the ablations, it seems like for layout generation, noising the discrete tokens faster than the continuous vectors gives better performance. This could be because denoising the bounding boxes faster allows the model to make the element type predictions better.

| Disc. Noise Schedule | Cont. Noise Schedule | Cont. Noise Steps | FID |
|---|---|---|---|
| $[0.5, 0.5, 0.5, 0.5]$ | $\text{lin}(0.0001, 0.02)$ | $[200, 200, 200, 200]$ | 13.19 |
| $[0.5, 0.5, 0.5, 0.5]$ | $\text{lin}(0.0001, 0.035)$ | $[200, 200, 200, 200]$ | 10.62 |
| $[0.5, 0.5, 0.5, 0.5]$ | $\text{cosine}(0.0001, 0.03)$ | $[200, 200, 200, 200]$ | 8.86 |
| $[0.5, 0.5, 0.5, 0.5]$ | $\text{cosine}(0.0001, 0.03)$ | $[100, 100, 300, 300]$ | 8.32 |
| $[0.5, 0.5, 0.5, 0.5]$ | $\text{cosine}(0.0001, 0.03)$ | $[25, 25, 50, 700]$ | 8.68 |
| $[0.5, 0.5, 0.5, 0.5]$ | $\text{cosine}(0.0001, 0.06)$ | $[10, 10, 10, 370]$ | 12.78 |
| $[0.75, 0.5, 0.25, 0.25]$ | $\text{cosine}(0.0001, 0.03)$ | $[10, 10, 10, 770]$ | 10.06 |
| $[0.5, 0.5, 0.5, 0.5]$ | $\text{cosine}(0.0001, 0.025)$ | $[10, 10, 10, 970]$ | 9.67 |
| $[0.5, 0.5, 0.5, 0.5]$ | $\text{cosine}(0.0001, 0.02)$ | $[10, 10, 10, 1170]$ | 10.83 |
| $[0.9, 0.75, 0.5, 0.5, 0.25]$ | $\text{cosine}(0.0001, 0.06)$ | $[50, 50, 50, 50, 50, 50]$ | 9.10 |
| $[0.5, 0.5, 0.5, 0.5, 0.5, 0.5]$ | $\text{cosine}(0.0001, 0.03)$ | $[10, 10, 10, 10, 10, 850]$ | 10.42 |
| $[0.99, 0.9, 0.8, 0.5, 0.25, 0.05]$ | $\text{cosine}(0.0001, 0.03)$ | $[400, 400, 70, 10, 10, 10]$ | 17.69 |

Table 11: Ablation on Discrete and Continuous Noise Schedules - PubLayNet

| Disc. Noise Schedule | Cont. Noise Schedule | Cont. Noise Steps | FID |
|---|---|---|---|
| $[0.5, 0.5, 0.5, 0.5]$ | $\text{cosine}(0.0001, 0.03)$ | $[10, 10, 10, 770]$ | 2.54 |
| $[0.5, 0.5, 0.5, 0.5]$ | $\text{cosine}(0.0001, 0.06)$ | $[10, 10, 10, 370]$ | 3.67 |
| $[0.5, 0.5, 0.5, 0.5]$ | $\text{cosine}(0.0001, 0.05)$ | $[10, 10, 10, 570]$ | 3.35 |
| $[0.5, 0.5, 0.5, 0.5]$ | $\text{cosine}(0.0001, 0.03)$ | $[300, 300, 100, 100]$ | 5.13 |
| $[0.5, 0.5, 0.5, 0.5]$ | $\text{cosine}(0.0001, 0.03)$ | $[100, 100, 300, 300]$ | 4.33 |
| $[0.9, 0.8, 0.7, 0.5, 0.5, 0.5]$ | $\text{cosine}(0.0001, 0.03)$ | $[10, 10, 10, 10, 380, 380]$ | 3.88 |

Table 12: Ablation on Discrete and Continuous Noise Schedules - RICO

**Sampling step comparisons:** Sampling time details for the baselines are given in Table 13. Recall that $I_1$ refers to the interleaving pattern $[t_1, p_1, \ldots, t_N, p_N]$ where $t_i$ is the (discrete) layout element type corresponding to the $i^{\text{th}}$ layout element and $p_i$ is the (continuous) bounding box positions/size of the $i^{\text{th}}$ layout element. $I_2$ refers to the interleaving pattern $[t_1, t_2 \ldots, t_N, p^c]$ where $t_i$ is again the (discrete) layout element type corresponding to the $i^{\text{th}}$ layout element and $p^c$ is a single continuous vector formed by concatenating the (continuous) bounding box positions/size of all layout elements.

| Model | Sampling steps per discrete token | Sampling steps per continuous vector | Total sampling steps |
|---|---|---|---|
| LayoutFlow | - | 50 | 50 |
| LayoutDM | 100 | - | 100 |
| DLT | 100 | 100 | 100 |
| LayoutDiffusion | 160 | - | 160 |
| IGD - $I_2$ interleaving | 4 | 800 | 880 |
| IGD - $I_1$ interleaving | 4 | 800 | 16080 |

Table 13: Sampling step comparison

For other baselines, since factorizability is assumed, sampling of all tokens/ vectors happen in parallel and hence, steps per element is identical to the total number of steps. However, this is not the case with IGD since sampling happens one element at a time - to see this, consider the IGD denoising process for 4 rounds. Each discrete element is sampled once per round - for layout generation with 20 discrete elements, this corresponds to 80 discrete sampling steps. Now, each continuous element is sampled multiple times per round - assuming we do 200 sampling steps per continuous vector per round (leading to 800 sampling steps per continuous

| Continuous Noise Schedule | Per-round Continuous Sampling Steps | Total Sampling Steps | Unconditioned FID | Class-conditioned FID | Class + Size conditioned FID |
|---|---|---|---|---|---|
| $\cos ine(0.0001, 0.55)$ | $[5, 5, 5, 20]$ | 780 | 4.69 | 1.18 | 0.92 |
| $\cos ine(0.0001, 0.43)$ | $[5, 5, 5, 35]$ | 1080 | 3.67 | 1.05 | 0.73 |
| $\cos ine(0.0001, 0.03)$ | $[10, 10, 10, 770]$ | 16080 | 2.54 | 1.06 | 0.96 |

Table 14: Ablation on Sampling Steps - RICO

| Continuous Noise Schedule | Per-round Continuous Sampling Steps | Total Sampling Steps | Unconditioned FID | Class-conditioned FID | Class + Size conditioned FID |
|---|---|---|---|---|---|
| $\cos ine(0.0001, 0.55)$ | $[5, 5, 5, 20]$ | 780 | 11.08 | 6.24 | 4.64 |
| $\cos ine(0.0001, 0.43)$ | $[5, 5, 20, 20]$ | 1080 | 10.20 | 4.32 | 2.69 |
| $\cos ine(0.0001, 0.03)$ | $[100, 100, 300, 300]$ | 16080 | 8.32 | 4.08 | 0.88 |

Table 15: Ablation on Sampling Steps - PubLayNet

vector across rounds), this leads to a total of 800 continuous sampling steps for $I_2$ and 16000 continuous sampling steps for $I_1$. The results reported in the main text use $I_1$ interleaving.

While IGD requires considerably higher number of sampling steps with $I_1$ interleaving, we believe it is justified due to the following reasons:

- Note that **none of the other baselines support denoising across continuous elements sequentially** - they **require** all continuous elements to be grouped into a single vector for DDPM style denoising. Hence, $I_1$ style denoising is **unique to our framework** - intuitively, $I_1$ allows denoising of one continuous element conditioned on other partially denoised continuous elements, which can help in better learning. In fact, recent work Wewer et al. (2025) has also demonstrated that performing diffusion on a sequence of image patches instead of the whole image drastically improves performance on MNIST Sudoku.

- Further, even if sampling steps are increased for baselines, they **do not show improvement** in performance: rather, the **performance saturates or even declines** as noted in Figure 3 of Levi et al. (2023) and Figure 8 of Guerreiro et al. (2025) with increasing sampling steps. IGD is **effectively able to make use of higher number of sampling steps** to improve performance *by utilizing the interleaving pattern $I_1$*, since the sampling steps are now split across 20 separate continuous vectors instead of a single vector as in other frameworks.

- Finally, we note that 800 sampling steps were chosen to maximize performance - reducing this can bring down the total number of sampling steps even with $I_1$ interleaving with the trade-off being slightly worse performance as demonstrated in the ablations given in Table 14 and 15.

**Best configuration:** We obtain the best results with the configuration in Table 16.

| Hyperparameter | PubLayNet | RICO |
|---|---|---|
| Interleaving Pattern/ Noise Order ($i_t$) | $I_1$ | $I_1$ |
| Discrete Noise Schedule ($\Pi_t$) | $[0.5, 0.5, 0.5, 0.5]$ | $[0.5, 0.5, 0.5, 0.5]$ |
| Continuous Noising Steps ($K_{i_t}^t$) | $[100, 100, 300, 300]$ | $[10, 10, 10, 770]$ |
| Continuous Noise Schedule ($\beta$) | $\cos ine(0.0001, 0.03)$ | $\cos ine(0.0001, 0.03)$ |
| Top-p | 0.99 | 0.99 |

Table 16: Best configuration for Layout Generation

### K.5 Compute Requirements:

All training was conducted on TPUv6e pods[Cloud], with each pod consisting of 8 TPU chips. For experiments on PubLayNet, models were trained on a single pod, with training times ranging upto 4 days. For RICO, models were were again trained on a single pod, with training times ranging upto 2 days. More information about the compute architecture and configuration at [link].

## L  Molecule Generation

### L.1  Other Baselines

While (Peng et al., 2023) and (Vignac et al., 2023b) are also diffusion based methods, they are not directly comparable. (Vignac et al., 2023b) proposes to generate 2D molecular graphs in tandem with 3D positions to allow better molecule generation. Our numbers cannot be directly compared with this work since they use a different list of allowed bonds, as well as use formal charge information. We also note that our framework can also be used to generate 2D molecular graphs along with 3D positions; we can also make use of the rEGNNs and uniform adaptive schedule proposed in (Vignac et al., 2023b). Hence, our framework can be thought of as complementary to (Vignac et al., 2023b). Similarly, (Peng et al., 2023) proposes to use the guidance of a bond predictor to improve molecule generation. Again, we cannot directly compare the numbers since they use a dedicated bond predictor to make bond predictions instead of a look-up table. The idea of bond predictor can also be incorporated in our framework seamlessly; hence our framework is again complementary to this work.

### L.2  Training Details

We train a Dis-Co DiT model with the following configuration:

| | |
|---|---|
| Number of Generalized DiT Blocks | 8 |
| Number of Heads | 8 |
| Model Dimension | 512 |
| MLP Dimension | 2048 |
| Time Embedding Input Dimension | 256 |
| Time Embedding Output Dimension | 128 |

Table 17: Model configuration for QM9

We use the AdamW optimizer (with $\beta_1 = 0.9$, $\beta_2 = 0.999$ and $\epsilon = 10^{-8}$) with no weight decay and with no dropout. We use EMA with decay 0.9999. We set the initial learning rate to 0 and warm it up linearly for 8000 iterations to a peak learning rate of $10^{-4}$; a cosine decay schedule is then applied to decay it to $10^{-6}$ over the training steps. For QM9, we train for 2.5 Million iterations with a batch size of 2048. We use pad tokens to pad the number of atoms to 29 if a molecule has fewer atoms.

**Distance-based embedding for atom positions:** We adapt the distance embedding part from the EGCL layer proposed in Hoogeboom et al. (2022). Consider a molecule with $N$ atoms; let us denote the atom position of the $i^{\text{th}}$ atom as $x_i$. Then, we begin by computing the pairwise distance between the $i^{\text{th}}$ atom and all the other atoms (including the $i^{\text{th}}$ atom itself) to get an $N-$ dimensional vector $d_i$. $d_i$ is fed into the Generalized DiT block and embedded to a vector of size $D$, where $D$ is the model dimension, using a linear projection. This $D$ dimensional array is processed as usual by the block and at the end of the block, it is projected back into an $N$ dimensional vector, which we call $m_i$, using another linear layer. Then, we modify $x_i$ as follows:

$$x_i \leftarrow x_i + \sum_{j \neq i} \frac{x_i - x_j}{d_{ij} + 1} m_{ij}$$

where $d_{ij}$ denotes the $j^{\text{th}}$ element of $d_i$ and $m_{ij}$ denotes the $j^{\text{th}}$ element of $m_i$. The distance $d_i$ is now recomputed using the modified $x_i$ and the process is repeated for each block. After the final block, we subtract out the initial value of $x_i$ from the output.

### L.3  Ablations

Unless specified otherwise, all the results reported in ablations use top-$p$ sampling with $p = 0.99$ and do not use the ReDeNoise algorithm at inference. For all molecule generation experiments, we noise the sequence in a round-robin fashion, and in each round, $\Pi(\phi)$ is constant for discrete tokens across all positions. Similarly,

$K_{i_t}^t$ which is the number of continuous noising steps per round, is constant across all positions per round. By default, we choose $\Pi(\phi)$ to be $[0.5, 0.5, 0.5, 0.5]$, where the 4 element sequence, which we refer to as the discrete noise schedule, denotes noising for 4 rounds with $\Pi(\phi)$ for the round chosen from the sequence. Similarly, the default value of $K_{i_t}^t$ is chosen to be $[200, 200, 200, 200]$, and we refer to this sequence as the continuous noising steps. Let us denote $\sum_t K_{i_t}^t$ as $K$. Note that $K$ is same across positions since we assume same number of continuous noising steps across positions per round. Given $K$, we use the following noise schedule for $\beta$:

$$\beta(j) = 0.03 + 0.5(0.0001 - 0.03)(1 + \cos\left(\left(\frac{j}{K}\right)\pi\right))$$

where $j$ is the total number of continuous noising steps at sequence time $t$ and element time $k$. We denote this noise schedule as cosine$(0.0001, 0.03)$.

**Interleaving pattern/ Noise order:** We broadly consider two interleaving patterns, similar to layout generation. In the first pattern, which we refer to as $I_1$, the atom positions of each atom is treated as a separate vector to form the interleaving pattern $[z_1, p_1, z_2, p_2, \ldots, z_n, p_n]$, where $z_i \in \mathbb{N}$ is the discrete atomic number and $p_i \in \mathbb{R}^3$ is its corresponding atom position. This interleaving pattern results in 29 discrete tokens and 29 continuous vectors. In the second pattern, which we refer to as $I_2$, the atom positions of all the $n$ atoms were bunched together as a single vector to form the interleaving pattern $[z_1, z_2, \ldots, z_n, p^c]$, where $p^c \in \mathbb{R}^{3n}$ is a single vector which is formed by concatenating the atom positions of all $n$ atoms. This interleaving pattern results in 29 discrete tokens and 1 continuous vector. The atom and molecule stability for these two configurations are given in Table 18.

| Interleaving Pattern | Atom. Stability | Mol. Stability |
|:---:|:---:|:---:|
| $I_1$ | 88.99 | 28.9 |
| $I_2$ | 98.07 | 83.83 |

Table 18: Ablation on Interleaving Pattern

As we can see, having the atom positions together helps improve performance by a large margin; we hypothesize that this could be because having the positions together allows the model to capture the symmetries of the molecules better. We choose the interleaving pattern $I_2$ for all further experiments.

**DDPM v/s DDIM:** We evaluate both DDPM and DDIM using the positions together interleaving pattern. The results are given in Table 19. DDPM outperforms DDIM by a large margin and hence we use DDPM for all experiments.

| Saampling Strategy | Atom. Stability | Mol. Stability |
|:---:|:---:|:---:|
| DDIM | 94.84 | 61.29 |
| DDPM | 98.07 | 83.83 |

Table 19: Ablation on Sampling Strategy

**Distance-based atom position embedding:** As we discussed in $L.2$, we use a distance-based embedding for the atom positions. We tried directly using the positions, as well as using both by concatenating distance along with the positions. The atom and molecule stability for these two configurations are given in Table 20.

As we can see, using the distance embedding leads to the best results. This could be due to the fact that molecules inherently have rotation symmetry, which distance-based embeddings capture more naturally. This could also be due to the fact that both atom and molecule stability are metrics which rely on the distance between atoms and allowing the model to focus on the distance allows it to perform better. Hence, we choose the distance-based atom position embedding for all further experiments.

| Embedding | Atom. Stability | Mol. Stability |
|---|---|---|
| Position | 91.87 | 55.93 |
| Distance | 98.07 | 83.83 |
| Position + Distance | 95.54 | 68.15 |

Table 20: Ablation on embedding

**Sequence time sampling:** While the sequence time $t$ is typically sampled uniformly between $0$ and $T - 1$, note that for the interleaving pattern with the positions together, only one sequence timestep per round corresponds to noising continuous vectors since we have $n$ discrete tokens and 1 continuous vector. This may make it slower for the model to learn the reverse process for the continuous vector. Hence, we also try a *balanced* sequence time sampling strategy, where we sample $t$ such that the time steps where continuous vector is noised is sampled with probability 0.5. For the same number of training steps, performance of both strategies are detailed in Table 21.

| Sequence Time Sampling | Atom. Stability | Mol. Stability |
|---|---|---|
| Uniform sampling | 97.92 | 79.78 |
| Balanced sampling | 98.24 | 84.47 |

Table 21: Ablation on Sequence Time Sampling

Since the balanced sampling strategy leads to better performance, we choose this strategy for all further experiments.

**Discrete noise schedule and continuous noising steps:** We fix the total number of noising rounds in the forward process as 4, the total number of continuous noising steps as 800 and the $\beta$ schedule as $\text{cosine}(0.0001, 0.03)$ based on initial experiments. The discrete noise schedule and continuous noising steps are then varied. Despite trying out multiple schedules, the default schedule of $[200, 200, 200, 200]$ and

| Discrete Noise Schedule | Continuous Noising Steps | Atom. Stability | Mol. Stability |
|---|---|---|---|
| $[0.5, 0.5, 0.5, 0.5]$ | $[200, 200, 200, 200]$ | 98.07 | 83.83 |
| $[0.5, 0.5, 0.5, 0.5]$ | $[100, 100, 300, 300]$ | 97.63 | 79.40 |
| $[0.5, 0.5, 0.5, 0.5]$ | $[300, 300, 100, 100]$ | 97.93 | 81.37 |
| $[0.5, 0.5, 0.5, 0.5]$ | $[100, 300, 100, 300]$ | 98.08 | 83.08 |
| $[0.75, 0.5, 0.5, 0.25]$ | $[100, 200, 200, 300]$ | 98.13 | 83.00 |
| $[0.85, 0.5, 0.5, 0.25]$ | $[50, 250, 200, 300]$ | 98.14 | 81.99 |

Table 22: Ablation on Noise Schedules

$[0.5, 0.5, 0.5, 0.5]$ give the best results; we use these noise schedules for further experiments. Results are given in Table 22.

**Effect of ReDeNoise:** We examine the effect of ReDeNoise algorithm at inference. Preliminary results indicated that noising and denoising for more than one round does not improve performance. Hence, we apply ReDeNoise for one round, but do multiple iterations of the noising and denoising. We observe the following: ReDeNoise improves performance upto 6 iterations, after which the metrics saturate. However, we see that there is a substantial improvement in the moelcular stability metric on using ReDeNoise. Table 23 gives the results of ReDeNoise in the unbalanced sequence time sampling setting. Since we observed performance improvement till 6 rounds, we used this for further experiments. The results for balanced sequence time sampling is given in Table 24.

| No. of times ReDeNoise is applied | Atom. Stability | Mol. Stability |
|:---:|:---:|:---:|
| No ReDeNoise | 97.94 | 80.24 |
| 1x | 98.23 | 83.42 |
| 2x | 98.37 | 85.17 |
| 3x | 98.46 | 85.78 |
| 4x | 98.48 | 86.20 |
| 5x | 98.52 | 86.49 |
| 6x | 98.60 | 87.11 |
| 7x | 98.48 | 86.30 |

Table 23: Ablation on ReDeNoise (unbalanced sequence time sampling)

| No. of times ReDeNoise is applied | Atom. Stability | Mol. Stability |
|:---:|:---:|:---:|
| No ReDeNoise | 98.24 | 84.47 |
| 6x | 98.74 | 89.46 |

Table 24: Ablation on ReDeNoise (balanced sequence time sampling)

**Effect of Top-p sampling:** We vary top-p sampling value at inference - results in Table 25.

| Top-p | Atom. Stability | Mol. Stability |
|:---:|:---:|:---:|
| 0.8 | 98.60 | 88.5 |
| 0.9 | 98.90 | 90.74 |
| 0.99 | 98.74 | 89.46 |

Table 25: Ablation on Top-p

**Sampling step comparisons:** Sampling time details are given in Table 26. Recall that $I_1$ refers to the interleaving pattern $[z_1, p_1, \ldots, z_N, p_N]$ where $z_i$ is the (discrete) atom type corresponding to the $i^{\text{th}}$ atom and $p_i$ is the (continuous) position of the $i^{\text{th}}$ atom. $I_2$ refers to the interleaving pattern $[z_1, z_2 \ldots, z_N, p^c]$ where $z_i$ is again the (discrete) atom type corresponding to the $i^{\text{th}}$ atom and $p^c$ is a single continuous vector formed by concatenating the (continuous) positions of all atoms.

| Model | Sampling steps per discrete token | Sampling steps per continuous vector | Total sampling steps |
|:---:|:---:|:---:|:---:|
| EDM | - | 1000 | 1000 |
| MuDiff | 1000 | 1000 | 1000 |
| IGD - $I_2$ interleaving | 4 | 800 | 916 |
| IGD - $I_1$ interleaving | 4 | 800 | 23316 |

Table 26: Sampling step comparison

Unlike layout generation, we obtain **best results in molecule generation with $I_2$ interleaving**. Hence, for this problem, **IGD requires less number of sampling steps as compared to baselines**. The choice of interleaving pattern ($I_1/I_2$) is hence a design choice dependent on the problem - the sampling step scaling varies accordingly from problem to problem.

**Best configuration:** After all the above ablations, we obtain the best results with the following configuration:

| | |
|---|---|
| Interleaving pattern/ Noise Order $(i_t)$ | $I_2$ |
| Atom Position Embedding | Distance-based |
| Sequence Time Sampling | Balanced |
| Discrete Noise Schedule $(\Pi_t)$ | $[0.5, 0.5, 0.5, 0.5]$ |
| Continuous Noising Steps $(K_{i_t}^t)$ | $[200, 200, 200, 200]$ |
| Continuous Noise Schedule $(\beta)$ | cosine$(0.0001, 0.03)$ |
| ReDeNoise | 6x |
| Top-p | 0.9 |

Table 27: Best configuration for QM9

## L.4 Compute Requirements:

All training was conducted on TPUv6e pods[Cloud], with each pod consisting of 8 TPU chips. For experiments on QM9, models were trained on two pods, with training times ranging upto 5 days. Experiments which varied batch sizes were done apart from the experiments reported - they were not reported since there wasn't any noticeable impact on the performance. More information about the compute architecture and configuration at [link].

# M  Tabular Data Generation

## M.1  Full Results

The performance of the IGD model on five tabular datasets is detailed in Table 28. The table reports key metrics including Wasserstein distances for training and test sets, coverage metrics, and F1 scores for generated and combined data, with the mean performance across all datasets provided in the last row.

Table 28: **Tabular Data Generation**: IGD Results on 5 Tabular Dataset

| Dataset | $W_{\text{train}} \downarrow$ | $W_{\text{test}} \downarrow$ | $\text{Coverage}_{\text{train}} \uparrow$ | $\text{Coverage}_{\text{test}} \uparrow$ | $F_1^{\text{gen}} \uparrow$ | $F_1^{\text{comb}} \uparrow$ |
|---|---|---|---|---|---|---|
| blood-transfusion | 0.163 | 0.250 | 1.000 | 0.993 | 0.599 | 0.600 |
| congress | 1.181 | 2.483 | 0.919 | 0.965 | 0.940 | 0.941 |
| car | 0.439 | 1.124 | 0.637 | 0.329 | 0.805 | 0.836 |
| tic-tac-toe | 0.822 | 1.971 | 0.662 | 0.323 | 0.921 | 0.937 |
| glass | 0.361 | 0.634 | 0.819 | 1.000 | 0.625 | 0.655 |
| Mean | 0.593 | 1.292 | 0.807 | 0.722 | 0.778 | 0.794 |

## M.2  Additional Metrics

We report results on $\text{coverage}_{tr}$ and $\text{coverage}_{te}$. These metrics measure the generation diversity computed between ($D_{tr}$ and $D^{gen}$) and ($D_{te}$ and $D^{gen}$) respectively. Results are given in Table 29.

Table 29: **Tabular Data Generation**: Results for additional metrics $\text{coverage}_{tr}$ and $\text{coverage}_{te}$

| Models | $\text{coverage}_{tr} \uparrow$ | $\text{coverage}_{te} \uparrow$ |
|---|---|---|
| HS3F-Euler | 0.788 | 0.671 |
| CS3F-Euler | 0.771 | **0.756** |
| HS3F-Rg4 | 0.804 | 0.661 |
| CS3F-Rg4 | 0.636 | 0.692 |
| ForestFlow | 0.700 | 0.735 |
| Ours | **0.807** | 0.722 |

## M.3  Training Details

We train a Dis-Co DiT model with the following configuration for all 5 datasets:

| | |
|---|---|
| Number of Generalized DiT Blocks | 4 |
| Number of Heads | 8 |
| Model Dimension | 512 |
| MLP Dimension | 2048 |
| Time Embedding Input Dimension | 256 |
| Time Embedding Output Dimension | 128 |

Table 30: Model configuration for Tabular Data Generation

We use the AdamW optimizer (with $\beta_1 = 0.9$, $\beta_2 = 0.999$ and $\epsilon = 10^{-8}$) with no weight decay and with no dropout. We use EMA with decay 0.9999. We set the initial learning rate to 0 and warm it up linearly for 8000 iterations to a peak learning rate of $7 \times 10^{-5}$; a cosine decay schedule is then applied to decay it to $10^{-6}$ over the training steps.

Since each of the 5 datasets have different number of samples, we use different batch sizes and training steps for each. These are described in 31.

| Dataset | Batch Size | Training Steps |
|---|---|---|
| blood-transfusion | 512 | 30000 |
| congress | 256 | 25000 |
| car | 1024 | 60000 |
| tic-tac-toe | 512 | 30000 |
| glass | 128 | 65000 |

Table 31: Batch sizes and training steps

Let $\{d_1, d_2, \ldots, d_{N_d}\}$ denote all the discrete tokens in the considered dataset and $\mathbf{c}$ denote the vector containing all continuous features. Further, $\{i_t\}$ is chosen to be the same across rounds and hence can equivalently be described by the noise order. Then, the IGD framework hyperparameters chosen for all 5 datasets is given in Table 32.

| | |
|---|---|
| Interleaving pattern/ Noise Order $(i_t)$ | $[d_1, d_2, \ldots, d_{N_d}, \mathbf{c}]$ |
| Number of Rounds $(r)$ | 4 |
| Discrete Noise Schedule $(\Pi_t)$ | $[0.5, 0.5, 0.5, 0.5]$ |
| Continuous Noising Steps $(K_{i_t}^t)$ | $[200, 200, 200, 200]$ |
| Continuous Noise Schedule $(\beta)$ | $\text{cosine}(0.0001, 0.03)$ |
| Top-p | 0.99 |

Table 32: Best configuration for Tabular Data Generation

## M.4 Compute Requirements:

All training was conducted on TPUv6e pods[Cloud], with each pod consisting of 8 TPU chips. For all datasets, experiments models were trained on a single pod, with training times ranging upto 5 hours. More information about the compute architecture and configuration at [link].

## N    Impact Statement

This paper contributes to the advancement of machine learning by introducing a novel constrained sampling algorithm. While generative machine learning models, particularly in text and image generation, have raised societal concerns, our work focuses on 3-SAT, molecule generation, layout generation, and tabular data generation, using publicly available datasets. At this stage, we foresee no direct societal impact. However, we acknowledge that future applications of our method in sensitive domains may necessitate a more thorough evaluation of potential societal implications.

## O    Licenses and Copyrights Across Assets

1. PubLayNet Benchmark

   - Citation: Zhong et al. (2019)
   - Asset Link: [link]
   - License: CDLA-Permissive-1.0, [link]

2. RICO Benchmark

   - Citation:  Deka et al. (2017)
   - Asset Link: [link]
   - License:Custom license agreement, [link]

3. QM9 Benchmark

   - Citation:  Ramakrishnan et al. (2014)
   - Asset Link: [link]
   - License: CC-BY 4.0, [link]

4. Random 3-SAT benchmark

   - Citation:  Ye et al. (2025)
   - Asset Link: [link]
   - License: Apache License 2.0, [link]

5. PySAT: Python toolkit for prototyping with SAT solvers

   - Citation:  Ignatiev et al. (2018)
   - Asset Link: [link]
   - License: MIT License, [link]

6. Tabular Datasets

   - Citation:  Kelly et al.
   - Asset Link: [link]
   - License: CC-BY 4.0, [link]

