# OpenReview forum: "Interleaved Gibbs Diffusion: Generating Discrete-Continuous Data with Implicit Constraints"
_TMLR — Under review for TMLR_

### Review · Reviewer_3q5C · 2026-04-03

**Summary Of Contributions:**

**Problem Statement**
Interleaved Gibbs Diffusion (IGD) is a diffusion-like generative framework for mixed discrete–continuous (DC) sequences that performs both forward noising and reverse denoising one element at a time (coordinate-wise updates), using a round-robin permutation over positions (the interleaving pattern). A key design claim is that sacrificing factorization in the reverse process enables exact reversal if the denoisers are ideal; the reverse is explicitly not factorizable.

**Strengths**
- The framework provides afirst DC diffusion setup that avoids the assumption of factorizability of the reverse denoising process. The avoidance of reverse factorization provides a significant improvement over previous joint DC generative models.

- The submission provides formal proofs showing that the forward process converges to a desired independent product noise distribution and that the reverse process exactly matches the target data distribution under ideal conditional denoisers.

- ReDeNoise as a predictor-corrector mechanism for diffusion models to mitigate approximation errors encountered during inference.

- State-space doubling is utilized to facilitate conditional sampling, so that the model can complete sequences even with partial constraints. Making the method even more useul under real-world use cases.

- State-of-the-art results or near enough are reported across four diverse domains: random 3-SAT (Boolean Satisfiability), document and user-interface layout generation, 3D molecule generation etc. The authors achieve these outcomes without highly specialized attention blocks or domain-specific architectures, emphasizing the technical superiority of their method.

**Weaknesses**:
- Decomposing factorizability in favor of Gibbs sampling where coordinates are processed sequentially, generating layouts requires a sequential pipeline that scales aggressively with the sequence length. While the results justify the cost for the $I_1$ scheme, the number of steps required seem exhorbitant for diminising returns.
- Another concern is the lack of a fixed recipe for the continuous and discrete noise schedules. While the tables in the appendix display many empirical combinations of continuous steps and schedules selected after experimental checks. A grid search would probably be required for real-world applicability. IGD’s own ablations on interleaving patterns show large metric swings when changing interleaving patterns and schedules (molecule stability changing dramatically), implying heavy tuning burden and reproducibility risk. This doesn't mean that the theory is wrong, but rather that it's fidgetty and the real-world applicability would be challenging.
- Gibbs sampling does break down at high dimensions. In large discrete or mixed spaces, the cost per effectively independent sample can become very large because autocorrelation grows and exploration remains local. Don't know how this can be addressed by the method, but there is a point I imagine when this method would break.

**Audience:**

Yes

**Audience Explanation:**

For sure, the method is general for discrete continuous data generation and shows results on a variety of datasets without any inductive bias in the architecture.

**Broader Impact Concerns:**

The work is broadly theoretical and while applications deliterious to humanity are possible with such a framework, that can't be addressed in a theoretical paper. No human data was collected as part of this paper and all experiments were done on existing datasets.

**Claims And Evidence:**

Yes

**Claims Explanation:**

- The mathematics is correct. The proofs are not particularly complex and are intuitive.
- The authors validate IGD across four diverse datasets containing highly coupled discrete and continuous constraints, outperforming or matching specialized domain-specific baselines without inductive bias or specialized architectures.

**Evidence:**
1. On random 3-SAT instances, switching to a non-factorized reverse process yields up to a $7.5\%$ accuracy improvement at $n=9$ variables compared to factorized diffusion baselines. On scaling up the parameters, near perfect satisfiability is achieved.
2. On the QM9 benchmark, the framework achieves a molecular stability rate of $90.5\%$, surpassing heavily tailored baselines like EDM and GeoLDM.
3. Similarly for the other datasets.

**Requested Changes:**

Not my area, so don't have much off the top of my mind to suggest. Would ask if performing multi-coordinate parallel updates or block-Gibbs sampling strategies where independent or weakly-coupled elements are updated simultaneously work in-practice, if not in theory? Could the authors see if that is indeed the case? Would speed up their method.

---

> ### Author Response · Authors · 2026-05-04
> **Factorizability, Noise Schedules and Relation to Gibbs Sampling**
>
> We thank the reviewer for the thorough review and providing a succinct summary of the main strengths of our work. We now address the main concerns raised by the reviewer:
>
> > Decomposing factorizability in favor of Gibbs sampling where coordinates are processed sequentially, generating layouts requires a sequential pipeline that scales aggressively with the sequence length. While the results justify the cost for the $I_1$ scheme, the number of steps required seem exorbitant for diminishing returns.
>
> We agree with the reviewer that we trade compute for quality in the case of layout generation. IGD was designed with the objective to optimize quality over efficiency. However, we would also like to note that we obtain better results with less compute for molecule generation. Therefore, this depends on the task as well as the choice of interleaving pattern. Further, ablations on layout generation (Table 14, 15) also reveal that IGD can do generation with substantially less compute, although with slightly worse quality.
>
> > Another concern is the lack of a fixed recipe for the continuous and discrete noise schedules. While the tables in the appendix display many empirical combinations of continuous steps and schedules selected after experimental checks. A grid search would probably be required for real-world applicability. IGD’s own ablations on interleaving patterns show large metric swings when changing interleaving patterns and schedules (molecule stability changing dramatically), implying heavy tuning burden and reproducibility risk. This doesn't mean that the theory is wrong, but rather that it's fidgetty and the real-world applicability would be challenging.
>
> We agree with the reviewer that the performance of IGD can be sensitive to the choice of hyperparameters. However, with respect to continuous and discrete noise schedules, we would also like to point out that a default of $[0.5, 0.5, 0.5, 0.5]$ discrete schedule and cosine$(0.0001, 0.03)$ seem to work well enough for all tasks. The choice of interleaving does seem to impact the performance; however, at least one of the two studied interleaving patterns ($I_1, I_2$) works well for all studied tasks. Hence, for a new task, we believe it would be sufficient to only ablate between $I_1$ and $I_2$ with this choice of noise schedule to obtain a good starting point.
>
> > Gibbs sampling does break down at high dimensions. In large discrete or mixed spaces, the cost per effectively independent sample can become very large because autocorrelation grows and exploration remains local. Don't know how this can be addressed by the method, but there is a point I imagine when this method would break.
>
> Gibbs sampling is typically used to generate samples from a  distribution whose log density is known (up to additive factors). This is usually a time-independent Markov Chain which can get stuck in metastable states, leading to large mixing times in high dimensions. These can be exponentially large (in dimension) in systems of interest. However, our setup is generative modeling, where we are given samples instead of log density. While our methods share similarities to Gibbs sampling, ours is a diffusion model which uses a _time-dependent_ Markov chain which exactly reverses a fast-mixing forward process (even in high dimensions). It is known in literature (and also shown specifically in our proof) that when the learning is accurate, number of steps is polynomial in dimension.
>
> > Would ask if performing multi-coordinate parallel updates or block-Gibbs sampling strategies where independent or weakly-coupled elements are updated simultaneously work in-practice, if not in theory? Could the authors see if that is indeed the case? Would speed up their method.
>
> It is definitely possible to modify our framework to incorporate block strategies where parallel updates happen within a block, as the reviewer suggested. However, this would break the exact reversal guarantee unless we know for sure that elements within a block are independent; arbitrary blocking can hence heavily degrade the quality. While an ``appropriate" blocking strategy could be very useful to obtain a tradeoff between computational cost and quality, we believe determining these blocks is non-trivial and leave exploration of such ideas to future work.

---

### Review · Reviewer_hRVK · 2026-04-16

**Summary Of Contributions:**

## Summary
The paper proposes a (discrete and continuous) denoising diffusion model named Interleaved Gibbs Diffusion (IGD), implementing the Gibbs sampling algorithm instead of the usual factorized sampling. The difference stands in that IGD proceeds by iteratively denoising only one dimension (or element) of the data at each time step, in a round-robin fashion, over many rounds. This denoises samples using the conditional distribution of the specific dimension given all other (unchanged) dimensions. This mechanism avoids the need to factorize the distribution of each denoising step (across dimensions of the data), which, as the authors state, might be unreasonable for the generation of data with strong implicit constraints. Additionally, the authors consider data comprising discrete and continuous dimensions. For discrete dimensions, the authors consider both a D3PM-style and a GGM-style model, while for continuous dimensions, the denoising/diffusion is implemented as a DDPM, both respecting the "one dimension at a time" principle. The authors evaluate the approach on the random 3-SAT problem, and layout, molecule, and tabular data generation datasets.

## Strengths
1. **Overall:** The idea of interleaving discrete and continuous data generation using a Gibbs-sampling scheme is interesting
2. **Writing:** The paper is, on a high level, well-structured
3. **Content:** The theory is sound, although I have not thoroughly checked every proof.

## Weaknesses
1. **Presentation:** I found the presentation of the work to be confusing at times.
	1. The concept of "implicit constraints" feels redundant, since the definition given by the authors, where the problem has constraints not known to the training algorithm/model, seems to align with the usual unconditional generative task. In this case, the constraints seem to simply be the "rules" of the dataset, which are actually present in any dataset (e.g., a dataset of images only containing cats). As opposed, constrained generative methods [1,2] integrate the constraints in the model, such that its support is constrained.
	2. The notation is sometimes confusing: $S^{(t)}$ is used in place of $\mathcal{S}^{(t)}$, which was introduced in the notation section. In addition, the time and dimension indices are inconsistently placed as superscripts and subscripts (for example, $x_t^{(n)}$ in Eq. (1), $s_{i_t}^{(t)}$ in other equations, and distributions $P_t$).
	3. Appendix A (with proofs) is never mentioned in the main text.
	4. MDM is not the correct name for the model in Ye et al., 2024.
2. **Methodology:**
	1. As the number of denoising steps $T$ increases linearly with respect to the dimensionality of data, it might grow larger than the factorized diffusion counterpart, where $T$ is a constant hyperparameter. This would make the proposed approach inefficient for very high-dimensional data such as images or large graphs.
	2. Lemma 3.1 assumes that $T\to\infty$. How does this differ from the assumption made for Eq.1? It seems that this also affects Lemma 3.2, where $\hat{S}^{(T)}$ is assumed to be sampled from $P_T$, but this distribution is only ever reached by $S^{(T)}$ for $T\to\infty$.
3. **Experiments:** the experimental section lacks some details necessary for the full picture. Furthermore, the molecular generation section presents some problems. My questions are:
	1. What do "category conditioned" and "category+size" conditioned mean in Layout generation, and how do you apply this specific conditioning to the model?
	2. In regard to molecule generation, do you represent edges in your data? From the definition in Section 6.2.1 only atoms are mentioned. Thus, how do you include edges in the architecture, and are they included in the Gibbs sampling? Additionally, how can you compare with DiGress which only generates 2D molecules? How do you obtain 3D positions in that case?
	3. In molecule generation, no distribution comparison metric is used (whereas layout uses FID and tabular uses 1-Wasserstein). Is there a reason you did not include any (e.g., Fréchet Chemnet Distance, FCD [3])?
	4. Please, mention all baselines in the main text. Did you train them, use checkpoints, or report results from the papers?
4. No code is provided

[1] Liu, G. H., Chen, T., Theodorou, E., & Tao, M. (2023). Mirror diffusion models for constrained and watermarked generation. _Advances in Neural Information Processing Systems_, _36_, 42898-42917.

[2] Christopher, J. K., Baek, S., & Fioretto, F. (2024). Constrained synthesis with projected diffusion models. _Advances in Neural Information Processing Systems_, _37_, 89307-89333.

[3] Preuer, K., Renz, P., Unterthiner, T., Hochreiter, S., & Klambauer, G. (2018). Fréchet ChemNet distance: a metric for generative models for molecules in drug discovery. _Journal of chemical information and modeling_, _58_(9), 1736-1741.

**Audience:**

Yes

**Audience Explanation:**

The paper proposes a diffusion generative model, which is at the current frontier of machine learning research.

**Claims And Evidence:**

No

**Claims Explanation:**

The evaluation of the model on layout and molecular datasets is not clear.

**Requested Changes:**

I advise the paper to be revised with the following **critical** changes:
1. Provide more clarity in the writing (see Weakness 1), and motivate the discussion of constraints in the setup of the paper (W1.1). I advise making the notation section more comprehensive
2. Fill in the theoretical gaps (W2), or discuss (in the rebuttal) whether these issues are not well-founded.
3. Include important material not presented in the main text, but included in the appendix. Too many useful details are included in the appendix (which I remark is at the discretion of the reviewers to be read). Some details, such as the reference to all baselines, whether they fit the setup, and details about the datasets that you are going to mention in the table, need to be included for the discussion to be meaningful. The main text should be stand-alone and allow reproducibility as much as possible. Also, consider referencing all appendices in the main text.
4. Revise or motivate the experiments as discussed in W3

**Minor revisions**:
1. Please, if possible, provide the link to an anonymous repository of the code
2. Some cited papers have been published, but are included as arxiv papers. A few examples:
	- Fishman et al. is published in TMLR,
	- Ye et al. 2024 is published in ICLR 2025
	- Varma et al., 2024, published in ICLR 2025

---

> ### Author Response · Authors · 2026-05-04
> **Regarding Methodology**
>
> We thank the reviewer for noting that the proposed idea is interesting and that the paper is well-structured on a high level. We also thank the reviewer for the detailed feedback regarding presentation, most of which we have tried to address.
>
> We now address the main concerns raised by the reviewer:
>
> > As the number of denoising steps $T$ increases linearly with respect to the dimensionality of data, it might grow larger than the factorized diffusion counterpart, where $T$ is a constant hyperparameter. This would make the proposed approach inefficient for very high-dimensional data such as images or large graphs.
>
> We agree with the reviewer that the number of denoising steps in IGD scales with the length of the sequence under consideration. However, we are particularly interested in problems where the most important objective is to model the strong dependencies between variables. For such tasks, although IGD might be more inefficient (computationally), there can be substantial quality gains, as can be seen from the reported empirical results.
>
> >  Lemma 3.1 assumes that $T \to \infty$. How does this differ from the assumption made for Eq.1? It seems that this also affects Lemma 3.2, where $\hat{S}^{(T)}$ is assumed to be sampled from $P_T$, but this distribution is only ever reached by $S^{(T)}$ for $T\to\infty$.
>
> We would like to first point out that the convergence in Lemma 3.1 is exponential. This is well known in literature for continuous diffusion (OU process converges exponentially) and we prove exponential convergence for the discrete part as well in the proof.
>
> Additionally, one can always choose the last step $\Pi_T(\phi)$ to be 1, so that all discrete elements are guaranteed to flip to random tokens in this step, ensuring convergence. Similarly, we can choose the continuous noise schedule to ensure that the last step results in the continuous vectors converging to pure noise. This ensures convergence in finite time and hence allows sampling from $P_T$ exactly.

---

> ### Author Response · Authors · 2026-05-04
> **Regarding Presentation**
>
> > The concept of "implicit constraints" feels redundant, since the definition given by the authors, where the problem has constraints not known to the training algorithm/model, seems to align with the usual unconditional generative task. In this case, the constraints seem to simply be the "rules" of the dataset, which are actually present in any dataset (e.g., a dataset of images only containing cats). As opposed, constrained generative methods [1,2] integrate the constraints in the model, such that its support is constrained.
>
> We agree with the reviewer’s point that implicit constraints refer to the "rules" of the dataset. We used this phrase only to bring attention to the fact that exact modeling of dependencies between variables can indeed help model these rules better and lead to better performance. We give concrete examples of these implicit constraints/ rules in Appendix B.
>
> We have specifically contrasted our definition with explicitly constrained generative methods in the introduction to avoid confusion.
>
> > The notation is sometimes confusing: $S^{(t)}$ is used in place of $\mathcal{S}^{(t)}$, which was introduced in the notation section. In addition, the time and dimension indices are inconsistently placed as superscripts and subscripts (for example, $x_t^{(n)}$ in Eq. (1), $s_{i_t}^{(t)}$ in other equations, and distributions $P_t$).
>
> We thank the reviewer for pointing this out. We have modified Eq (1) (and related text) to use notation consistent with the rest of the paper (with time appearing as a superscript) in the revised version. For distributions, since we do not have a position index and only have time, we prefer to use a subscript as this is standard in literature.
>
> In the notation section, we use $\mathcal{S}$ to refer to the state space itself. $S$ denotes a random variable/sequence whose realization is an element of this state space. We believe this is consistent with the exposition throughout the paper; if there is any specific inconsistency, we will be happy to fix it.
>
> > Appendix A (with proofs) is never mentioned in the main text.
>
> We have now added this immediately after Lemma 3.1.
>
> > MDM is not the correct name for the model in Ye et al., 2024.
>
> The name was revised from MDM to MGDM by Ye et al. in a revision. We have changed this in the revised version.
>
> > Include important material not presented in the main text, but included in the appendix. Too many useful details are included in the appendix (which I remark is at the discretion of the reviewers to be read). Some details, such as the reference to all baselines, whether they fit the setup, and details about the datasets that you are going to mention in the table, need to be included for the discussion to be meaningful. The main text should be stand-alone and allow reproducibility as much as possible. Also, consider referencing all appendices in the main text.
>
> We have moved descriptions of baselines from Appendix to main text for all experiments in the revised version. We have also referenced all sections of the Appendix in the main text.
>
>
> > Some cited papers have been published, but are included as arxiv papers.
>
> Thanks for pointing this out. We have revised arXiv citations to the published versions wherever applicable.

---

> ### Author Response · Authors · 2026-05-04
> **Regarding Experiments**
>
> > What do "category conditioned" and "category+size" conditioned mean in Layout generation, and how do you apply this specific conditioning to the model?
>
> Category conditioned means that the types of elements (i.e., the categories) of the elements in the layout to be generated are pre-determined. The requirement is to generate the positions and sizes of these elements. "Category+size" conditioned means that the types as well as sizes of the elements are pre-determined. The requirement is only to position these elements appropriately. We have added a more detailed description in the layout generation section of the revised version.
>
> We use the exact strategy explained in Section 3.3 for conditioning. In summary, we use a binary mask to indicate parts of the sequence that are fixed/ conditioned on and feed in this mask to the neural network. During forward and reverse processes, parts which are conditioned on are unchanged.
>
> > In regard to molecule generation, do you represent edges in your data? From the definition in Section 6.2.1 only atoms are mentioned. Thus, how do you include edges in the architecture, and are they included in the Gibbs sampling? Additionally, how can you compare with DiGress which only generates 2D molecules? How do you obtain 3D positions in that case?
>
>
> We only generate atom types and positions: edges are then inferred from a look-up table by computing the distance between atoms (The exact same setup as the cited MuDiff (Hua et. al.)).
>
> All baseline metrics were also taken from MuDiff, as mentioned in the caption of Table 3.  Even though DiGress only generates 2D molecules, one can still compute the metrics that we have reported without explicitly computing the 3D positions.
>
> > In molecule generation, no distribution comparison metric is used (whereas layout uses FID and tabular uses 1-Wasserstein). Is there a reason you did not include any (e.g., Fréchet Chemnet Distance, FCD [3])?
>
> > Please, mention all baselines in the main text. Did you train them, use checkpoints, or report results from the papers?
>
> As mentioned above as well as in the caption for Table 3, we take all baseline metrics from MuDiff. We have cited the papers from which the baselines metrics were taken in the captions of all Tables. Since we did not re-train or evaluate the baselines, we only reported metrics which were already available in common across all baselines (and hence did not include additional metrics like FCD). We have moved description of baselines from the Appendix to the main text in the revision.

---

> > ### Comment · Reviewer_hRVK · 2026-05-26
> >
> > Dear authors,
> > Thank you for the thorough answer.
> >
> > > We agree with the reviewer that the number of denoising steps in IGD scales with the length of the sequence under consideration. However, we are particularly interested in problems where the most important objective is to model the strong dependencies between variables. For such tasks, although IGD might be more inefficient (computationally), there can be substantial quality gains, as can be seen from the reported empirical results.
> >
> > I would argue that most, if not all generation problems rely on modeling the strong dependencies between variables (e.g., among pixels in images or among nodes in a graph). Can you clarify what the specific target problems are or, if the problem is more general, what properties of IGD are desired as opposed to standard diffusion, as both are universal approximators of probability distributions?
> >
> > > Additionally, one can always choose the last step $\Pi_T(\phi)$ to be 1, so that all discrete elements are guaranteed to flip to random tokens in this step, ensuring convergence. Similarly, we can choose the continuous noise schedule to ensure that the last step results in the continuous vectors converging to pure noise. This ensures convergence in finite time and hence allows sampling from  $P_T$ exactly.
> >
> > This satisfies my doubt, but raises another: isn't this the same as in standard discrete diffusion [1], where $x_t$ converges due to how $Q_t$ is defined, e.g., leading to a uniform distribution?
> >
> > > We only generate atom types and positions: edges are then inferred from a look-up table by computing the distance between atoms (The exact same setup as the cited MuDiff (Hua et. al.)).
> >
> > I checked the paper, but I could not find the setup where edges are inferred from distances. Could you point it out? Also, how do you define such a look-up table for bond types? It is known that there is overlap in bond length ranges across bond types (e.g., in [2]).
> >
> > > All baseline metrics were also taken from MuDiff, as mentioned in the caption of Table 3. Even though DiGress only generates 2D molecules, one can still compute the metrics that we have reported without explicitly computing the 3D positions.
> >
> > I agree that metrics can still be computed, but missing values are filled by RDkit. Then, the task becomes fundamentally different, as DiGress cannot learn 3D coordinates. Wouldn't MiDi [3] be a better comparison in place of DiGress? I saw in appendix L.1 that you discuss this aspect but the missing comparison weakens the empirical results. For example, you might consider a version of MiDi where formal charges are not considered.
> >
> > > Since we did not re-train or evaluate the baselines, we only reported metrics which were already available in common across all baselines (and hence did not include additional metrics like FCD).
> >
> > For a generative task, I would argue that not including at least one metric comparing the distribution of generated data and test data does not provide the full picture. This weakens the evaluation of molecule generation on the probability estimation side. If this is not the aim, please highlight it in the manuscript.
> >
> > You can consider all other raised points as solved.
> >
> > [1] Austin, J., Johnson, D. D., Ho, J., Tarlow, D., & Van Den Berg, R. (2021). Structured denoising diffusion models in discrete state-spaces. _Advances in neural information processing systems_, _34_, 17981-17993.
> >
> > [2] Cognolato, S., Rigoni, D., Ballarini, M., Serafini, L., Moro, S., & Sperduti, A. (2026). D4: Distance diffusion for a truly equivariant molecular design. _Neurocomputing_, _679_, 133197. doi:10.1016/j.neucom.2026.133197
> >
> > [3] Vignac, C., Osman, N., Toni, L., & Frossard, P. (2023, September). Midi: Mixed graph and 3d denoising diffusion for molecule generation. In _Joint European Conference on Machine Learning and Knowledge Discovery in Databases_(pp. 560-576). Cham: Springer Nature Switzerland.

---

> > > ### Author Response · Authors · 2026-06-10
> > > **Further Clarifications**
> > >
> > > We thank the reviewer for the continued engagement and constructive comments. We now address the further questions raised by the reviewer:
> > >
> > > > I would argue that most, if not all generation problems rely on modeling the strong dependencies between variables (e.g., among pixels in images or among nodes in a graph). Can you clarify what the specific target problems are or, if the problem is more general, what properties of IGD are desired as opposed to standard diffusion, as both are universal approximators of probability distributions?
> > >
> > > We agree with the reviewer that all generation problems involve modeling *some* dependencies between variables. We have explained this exact point in the introduction, where we use this point to raise the question: how well does a generative algorithm model these dependencies?  Standard discrete diffusion models assume factorizability in each denoising step. If the number of steps is finite, these models, regardless of perfect training, cannot model all dependencies exactly. In contrast, IGD, if trained perfectly, can model these dependencies exactly in finite steps. This gives IGD capability to model constraints that models which assume factorizability cannot. For problems where this capability is desirable, IGD can achieve superior performance, as we empirically demonstrate.
> > >
> > > To illustrate this point, we consider a toy example in two dimensions. Consider two random variables $x_0, x_1$ such that $x_0 = x_1$ almost surely and $x_0, x_1 \in \{0,1\}$. Start our diffusion model with $x_0^{(0)}, x_1^{(0)}$ being Ber(1/2) random variables. Then, flipping them via a factorized diffusion model will always result in the wrong distribution since each token is flipped independently. This results in constraint violation. However, if we flip them one by one based on Gibbs-style sampling, we can satisfy the constraint $x_0 = x_1$.
> > >
> > >
> > > To conclude, for certain problems (language generation for instance), one may be able to obtain a reasonable performance by modeling these dependencies only approximately. Hence, discrete diffusion models, which assume factorizability can achieve good quality in very few steps. However, for problems like 3-SAT, where each variable is tightly coupled to other variables, exactly modeling these constraints (which IGD does) can improve performance.
> > >
> > > > This satisfies my doubt, but raises another: isn't this the same as in standard discrete diffusion [1], where $x_t$ converges due to how $Q_t$ is defined, e.g., leading to a uniform distribution?
> > >
> > >
> > > The reviewer is right in pointing out that our method is similar to standard discrete diffusion in the sense that as $T \rightarrow \infty$, the distribution tends to a uniform distribution. The main structural difference between our method and standard discrete diffusion is that we only flip/noise one element independently at each timestep during both forward and reverse processes, whereas in other methods, one typically noises/denoises multiple elements at once assuming a product distribution.
> > >
> > > > I checked the paper, but I could not find the setup where edges are inferred from distances. Could you point it out? Also, how do you define such a look-up table for bond types? It is known that there is overlap in bond length ranges across bond types (e.g., in [2]).
> > >
> > > We apologize for the confusion; the exact details are given in the paper arxiv:2203.17003 (the paper we referred to in the previous response cites this work for its setup). The look-up tables are defined in Tables 7, 8 and 9 in Appendix C of this paper. This setup is standard in the molecule generation literature and is followed across most of the baselines. As the reviewer suggested, this approach may not be entirely accurate since there could be overlaps in bond length ranges.

---

> > > > ### Author Response · Authors · 2026-06-10
> > > > **Further Clarifications (Ctd.)**
> > > >
> > > > > I agree that metrics can still be computed, but missing values are filled by RDkit. Then, the task becomes fundamentally different, as DiGress cannot learn 3D coordinates. Wouldn't MiDi [3] be a better comparison in place of DiGress? I saw in appendix L.1 that you discuss this aspect but the missing comparison weakens the empirical results. For example, you might consider a version of MiDi where formal charges are not considered.
> > > >
> > > > We first note that for atom stability and molecule stability, one only looks at the valency of atoms; hence the edges generated by DiGress can directly be used to compute this valency without having to compute any other missing values.
> > > >
> > > > While we agree that a comparison with MiDi under identical conditions can be helpful, this would either require adapting their setup to generate only atom types and positions or our setup to generate edges (bonds between atoms) explicitly rather than implicitly as we do currently. We believe this involves significant changes to our work in terms of algorithms and modeling, which is suitable for a follow-up paper given its complexity.
> > > >
> > > > > For a generative task, I would argue that not including at least one metric comparing the distribution of generated data and test data does not provide the full picture. This weakens the evaluation of molecule generation on the probability estimation side. If this is not the aim, please highlight it in the manuscript.
> > > >
> > > > While we have not included metrics like FCD, we note that atom stability and molecule stability are the main metrics reported across all prior works considered. Evaluating these would allow a direct comparison. Further, the majority of the baselines also do not report metrics like FCD, making it difficult to contextualize our work without rerunning these baselines.
> > > >
> > > > Evaluating the metrics considered allows us to check whether the implicit constraints are being satisfied, which is the overarching goal of the paper. We also note that the metrics such as a stability and molecule stability of the model-generated dataset are close to that of the test dataset.

---

### Review · Reviewer_iSGZ · 2026-04-25

**Summary Of Contributions:**

This paper introduces Interleaved Gibbs Diffusion (IGD), a novel generative modeling framework for discrete-continuous data that addresses a fundamental limitation in existing diffusion models. The authors argue that current discrete and discrete-continuous diffusion models assume factorizable denoising distributions, which is problematic for generation tasks with strong implicit constraints where variables are highly interdependent. To motivate their approach, they first demonstrate empirically on 3-SAT problems that non-factorized Gibbs-sampling style diffusion outperforms factorized methods by up to 7%, with larger improvements as constraints increase. The main contribution is the IGD framework itself, which extends time-dependent Gibbs sampling to handle both discrete and continuous elements while provably achieving exact reversal with ideal denoisers, without assuming factorization. The authors show that these ideal denoisers can be trained using well-understood objectives: classification for discrete elements and conditional score matching for continuous ones. The framework also supports conditional generation via state-space doubling and inference-time refinement through a ReDeNoise algorithm. Empirically, IGD achieves state-of-the-art results on molecule generation, layout generation, and tabular data generation without relying on domain-specific architectures or auxiliary losses, with comprehensive ablation studies provided for each task.

**Additional Comments:**

I think the methodology has potential, but the paper lacks clarity, especially for readers who are not experts in the field. Adapting the processes to unsuitable data is key and should be addressed with greater precision and rigor.

**Audience:**

Yes

**Audience Explanation:**

The problem addressed by this paper is highly relevant to a broad segment of TMLR's audience. Generative modeling with implicit constraints is a fundamental challenge encountered across numerous contemporary applications, including molecular design, layout generation, tabular data synthesis, and beyond. The limitation of factorization assumptions in existing diffusion models is a recognized bottleneck that affects many practitioners working with structured or constrained data.
The paper's findings appear promising at first glance, demonstrating state-of-the-art results across multiple diverse domains. Importantly, the proposed IGD framework is designed to be general-purpose, applicable to various types of discrete-continuous data without requiring domain-specific architectural modifications or auxiliary losses. This universality makes the approach particularly valuable, as it could potentially be adopted by researchers working across different application domains who face similar challenges with constrained generation.
Furthermore, the theoretical contribution, establishing a discrete-continuous diffusion framework that avoids factorization while guaranteeing exact reversal, addresses a gap in the literature that is of interest to both theoreticians and practitioners in the generative modeling community.

**Claims And Evidence:**

No

**Claims Explanation:**

While the paper presents convincing empirical results and a rigorous theoretical framework, the presentation significantly hinders accessibility and clarity, particularly for non-specialist readers. The paper is dense with mathematical notation, and key concepts are insufficiently contextualized or explained intuitively.
A major conceptual challenge is the adaptation of techniques originally designed for temporal stochastic processes to non-temporal data. The sequential ordering of variables imposed by the Gibbs sampling framework is not immediately intuitive in this context, particularly since there is no underlying causal structure among the variables being generated. The authors do not adequately motivate or explain why sequencing variables (the "noise order") is appropriate for problems like molecule or layout generation, where no natural temporal or causal ordering exists. More concrete examples or visual illustrations demonstrating how this sequential processing captures dependencies would greatly improve understanding.
Despite these presentation issues, the core approach appears sound and addresses an important problem: generating data while respecting implicit constraints without factorization assumptions. The empirical evidence across multiple domains (3-SAT, molecules, layouts, tabular data) is comprehensive and demonstrates clear improvements. However, the lack of clarity in exposition may limit the paper's impact and reproducibility.

**Requested Changes:**

- **2 Benefits of Exact Reversal for Problems with Implicit Constraints**:
  1. *let $x_s = (x_s^{(1)}, \cdots, x_s^{(N)})$* : define $N$.
  2. *In practice, for finite timesteps, equation 1 does not hold and hence the reverse process is not exact.* Explain why
  3. The third paragraph (illustration with SAT problem) is not clear. Is this a discrete or continuous case?
  4. Table1: Accuracy is a poor metric if the data is imbalanced. It would be advisable to display the distribution of the predicted variable and/or choose a more robust metric (such as the F1 score).
- **3 Interleaved Gibbs Diffusion**:
  1. $L = L_1 + L_2$ define $L_1$ and $L_2$.
  2. $\mathbb{N} \cup \{0\}$ ? But isn't {0} already in $\mathbb{N}$?
  3. **3.1 Forward Process**:
    - a. The process of adapting to non-temporal data should be better explained
    - b. How is $s^{(0)}$ defined for data that is not indexed?
    - c. *Noise Order*: I think this step is crucial because it allows processes to be adapted to the data. I’m not convinced that a random order should be used in sequential learning because the order of causality is unknown. A causal analysis (graph) could be helpful in this regard. Even when repeated, a random approach does not seem to guarantee the robustness of the generation and the interdependencies.
    - d. How do you choose $r$ in practice?
    - e. How is the distribution $\Pi$ defined?
    - f. What does $z_t \ne \phi$ represent?
    - g. The continuous form is really unclear
  4. **3.2 Reverse Denoising Process**:
    - a. Why is the indexing of the probability mass function denoted by $S_{-i_t}^{(t)} = s_{-i_t}$, but not in the continuous case?
    - b. *ReDeNoise*: This part is really unclear to me.
    - c. I don't find Algorithm 1 to be very useful. I suggest including a diagram or an algorithm for the global approach using the forward process.
  5. **Conditional Generation**: Providing an example of how it can be used would help highlight the value of this feature
- **6. Experiments**: This section is really clear, well-organized, and a pleasure to read
  1. Tables 1 & 3 : Uncertainty measurements are missing
  2. The competitors are not listed
  3. The experimental protocols are not clearly defined
  4. **6.3 Tabular data generation**:
    - a. Extending this approach to mixed data, i.e., data with categorical variables (which is very common in practice), would be a real advantage.
    - b. Why aren't well-known generators like CTGAN, TVAE, etc., included in the competition?
    - c. $D_{gen}$ is preferred over $D^{gen}$ to ensure consistency with $D_{tr}$

---

> ### Author Response · Authors · 2026-05-04
> **Regarding Sequential Ordering**
>
> We thank the reviewer for pointing out that IGD is a general framework which does not require domain-specific modifications and that the theoretical contribution addresses a gap in the literature. We also appreciate the detailed comments regarding presentation; we have tried our best to accommodate these in the revised version.
>
> The primary concern of the reviewer seems to be that IGD imposes a temporal/ sequential ordering which may not be naturally present in the data. To address this, we now point to a few facts:
>
> - First, irrespective of whether the variables in the data have a natural sequencing order or not, IGD is guaranteed to recover the exact distribution, as long as learning is perfect. The chosen noise order only impacts the difficulty of learning/ sample complexity. Hence, we do not believe that the following is a concern:
>
> > Even when repeated, a random approach does not seem to guarantee the robustness of the generation and the interdependencies.
>
> - Second, as the reviewer suggested, unlike tasks like language learning, there is no clear method/ strategy to obtain a causal ordering for data like layouts and molecules. Obtaining the order of causality for such problems is highly non-trivial and beyond the scope of this work. The _main claim of our work is that IGD can still work well despite the ordering being random_. As a parallel, we point to the fact that diffusion language models (which are not causal and sample in random order) are competitive to autoregressive language models which impose causality.
>
> - Third, the fact that IGD does well _empirically_ on multiple tasks without natural causal ordering is evidence in itself that IGD with random ordering is effective.  Another way to think about it is that, due to there being multiple rounds during sampling with bi-directional attention, it is possible for the model to infer arbitrary dependencies through backtracking. This is most evident from the 3-SAT example - while autoregressive models, which impose causal ordering with unidirectional/ causal attention struggle in this problem, GGM/IGD achieves excellent performance.
>
> To conclude, the introduced noise order is purely a mechanism to enable exact reversal. It does not assume or force any causal dependencies between variables in the data at a fundamental level.

---

> ### Author Response · Authors · 2026-05-04
> **Regarding Presentation**
>
> > let $x_s = (x_s^{(1)}, \cdots, x_s^{(N)})$ : define $N$.
>
> > $L = L_1 + L_2$ define $L_1$ and $L_2$.
>
> > $D_{gen}$ is preferred over $D^{gen}$ to ensure consistency with $D_{tr}$
>
> > The competitors are not listed
>
> These have been addressed in the revision.
>
> > In practice, for finite timesteps, equation 1 does not hold and hence the reverse process is not exact. Explain why
>
> This is explained in detail in the work already cited in the paper (arxiv:2406.04329). In summary, the true distribution is a joint distribution across all tokens/ variables. Assuming this is a product distribution is an approximation and is exact only in the infinitesimal limit. This is a well-known problem in discrete diffusion literature where aggressive parallel denoising with product distribution assumption leads to performance degradation. In fact, addressing this issue is an active area of research and many concurrent works (such as arxiv:2509.22738) also explore solutions to this problem.
>
> > The third paragraph (illustration with SAT problem) is not clear. Is this a discrete or continuous case?
>
> The SAT problem is a discrete case; hence we use autoregressive models and discrete diffusion models as baselines. This is just an illustration to show why exact modeling of the joint distribution can lead to performance gains.
>
> > Table1: Accuracy is a poor metric if the data is imbalanced. It would be advisable to display the distribution of the predicted variable and/or choose a more robust metric (such as the F1 score).
>
> It is not entirely clear to us what the reviewer is referring to here. As described in Appendix J, we only consider SAT problems for which solutions exist. The only objective of the model is to predict the solution to such a SAT problem. In such a setup, we believe accuracy is indeed a good metric. We also note that the setup exactly mirrors prior work.
>
> >  $\mathbb{N} \cup {0}$ ? But isn't {0} already in $\mathbb{N}$?
>
> As far as we are aware, 0 is not typically included in the set of natural numbers in most mathematical textbooks (see for instance: Introduction to Real Analysis, Bartle and Sherbert).
>
> > How is $s^{(0)}$ defined for data that is not indexed?
>
> For tuples where each entry is a real number or a categorical variable, one can always assign some indexing. In fact, different choices of indexing are explored as different interleaving patterns in ablations.
>
> > How do you choose $r$ in practice?
>
> In general, this is a hyperparameter. The only hard requirement is that $r >1$. In practice, we have found that $r = 4$ is a good choice.
>
> > How is the distribution $\Pi$ defined?
>
> As discussed in the paper, $\Pi_t$ can be any distribution over the discrete token space (with the additional token $\phi$ included). Different choices of $\Pi$ for which we did ablations are discussed in the Appendices.
>
> > What does $z_t \ne \phi$ represent?
>
> $z_t$ is sampled according to $\Pi_t$. When $z_t = \phi$, the current discrete token in the sequence remains unchanged. If $z_t \ne \phi$, i.e., $z_t$ is a token from the vocabulary other than $\phi$, then the current discrete token is flipped to $z_t$. Together, these represent the flipping mechanism fundamental to forward process of the discrete tokens.
>
> > The continuous form is really unclear
>
> Could the reviewer clarify what exactly is unclear?

---

> ### Author Response · Authors · 2026-05-04
> **Regarding Presentation (ctd.)**
>
> > Why is the indexing of the probability mass function denoted by $S_{-i_t}^{(t)} = s_{-i_t}$, but not in the continuous case?
>
> Note that even for the discrete case, we have provided both alternatives. Theoretically both can lead to exact reversal.  For continuous vectors, we only explore the alternative without the indexing, because this allows us to continue denoising from partially denoised vectors. The other alternative would require starting denoising again from pure Gaussian noise which may be computationally suboptimal considering that the continuous vectors have a notion of continuity across denoising rounds.
>
> > _ReDeNoise_: This part is really unclear to me.
>
> As noted by Reviewer 3q5C, ReDeNoise is an adaptation of standard predictor-corrector techniques to our setup. These are well-known diffusion sampling algorithm modifications to improve quality. We have provided a detailed mathematical exposition of this algorithm in Appendix C.
>
> > I don't find Algorithm 1 to be very useful. I suggest including a diagram or an algorithm for the global approach using the forward process.
>
> We have illustrated the forward process in Figure 1. We believe this combined with Algorithm 1 provides a comprehensive exposition of our technique. If the reviewer has specific suggestions for improvement, we would be happy to include them.
>
> > **Conditional Generation**: Providing an example of how it can be used would help highlight the value of this feature
>
> We have used conditional generation in the layout generation tasks - specifically "category conditioned" and "category  + size conditioned" generation.
>
> > Tables 1 & 3 : Uncertainty measurements are missing
>
> As discussed above, for Table 1, accuracy is the only relevant metric since we only consider SAT problems for which at least one solution exists. In Table 3, for molecule generation, we have reported the mean and variance of metrics computed on IGD generations (the baseline metrics were taken from prior literature). For Tabular Data generation, we do report 1-Wasserstein distance, which is a distributional metric.
>
> >  The experimental protocols are not clearly defined
>
> We have cited the relevant works whose experimental protocol we follow for each experiment. We have provided further implementational details in the Appendices.
>
> > Extending this approach to mixed data, i.e., data with categorical variables (which is very common in practice), would be a real advantage.
>
> All the five considered datasets have categorical variables.
>
> > Why aren't well-known generators like CTGAN, TVAE, etc., included in the competition?
>
> We benchmarked against existing diffusion/flow based methods on the same datasets and same setup as the cited prior work. Since we did not run other methods ourselves, a direct comparison was not possible since these generators were originally evaluated on different datasets.